# Effects of complex terrain on the shortwave radiative balance: a sub-grid-scale parameterization for the GFDL Earth System Model version 4.1

Enrico Zorzetto[1], Sergey Malyshev[2], Nathaniel Chaney[3], David Paynter[2], Raymond Menzel[2,4], and Elena Shevliakova[2]

[1]Program in Atmospheric and Oceanic Sciences, Princeton University, Princeton, NJ, USA
[2]NOAA OAR Geophysical Fluid Dynamics Laboratory, Princeton, NJ, USA
[3]Department of Civil and Environmental Engineering, Duke University, Durham, NC, USA
[4]University Corporation for Atmospheric Research, Princeton, NJ, USA

**Correspondence:** Enrico Zorzetto (ez6263@princeton.edu)

**Abstract.** Parameterizing incident solar radiation over complex topography regions in Earth system models (ESMs) remains a challenging task. In ESMs, downward solar radiative fluxes at the surface are typically computed using plane-parallel radiative transfer schemes, which do not explicitly account for the effects of a three-dimensional topography, such as shading and reflections. To improve the representation of these processes, we introduce and test a parameterization of radiation–topography interactions tailored to the Geophysical Fluid Dynamics Laboratory (GFDL) ESM land model. The approach presented here builds on an existing correction scheme for direct, diffuse, and reflected solar irradiance terms over three-dimensional terrain. Here we combine this correction with a novel hierarchical multivariate clustering algorithm that explicitly describes the spatially varying downward irradiance over mountainous terrain. Based on a high-resolution digital elevation model, this combined method first defines a set of sub-grid land units ("tiles") by clustering together sites characterized by similar terrain–radiation interactions (e.g., areas with similar slope orientation, terrain, and sky view factors). Then, based on terrain parameters characteristic for each tile, correction terms are computed to account for the effects of local 3D topography on shortwave radiation over each land unit. We develop and test this procedure based on a set of Monte Carlo ray-tracing simulations approximating the true radiative transfer process over three-dimensional topography. Domains located in three distinct geographic regions (Alps, Andes, and Himalaya) are included in this study to allow for independent testing of the methodology over surfaces with differing topographic features. We find that accounting for the sub-grid spatial variability of solar irradiance originating from interactions with complex topography is important as these effects led to significant local differences with respect to the plane-parallel case, as well as with respect to grid-cell-scale average topographic corrections. We further quantify the importance of the topographic correction for a varying number of terrain clusters and for different radiation terms (direct, diffuse, and reflected radiative fluxes) in order to inform the application of this methodology in different ESMs with varying sub-grid tile structure. We find that even a limited number of sub-grid units such as 10 can lead to recovering more than 60 % of the spatial variability of solar irradiance over a mountainous area.

## 1 Introduction

The presence of three-dimensional topography exerts an important control on the amount of solar radiation received by land. Over complex terrain, the incoming solar beam not only undergoes scattering and absorption within the atmospheric column but is also further modulated by the relative orientation of land surfaces and the potential shading and reflection

effects of neighboring slopes (Sirguey, 2009; Lenot et al., 2009; Lamare et al., 2020; Picard et al., 2020). The effect of surface roughness was recently shown to have important effects over snow-covered surfaces, leading to a net decrease
in surface reflectivity (Larue et al., 2020).

Together, these effects lead to a spatially heterogeneous distribution of the radiative fluxes received by the surface. In turn, this heterogeneity can have important consequences for the local energy and water balance and interact with other
spatially varying land processes such as evaporation (Brutsaert, 2013), snow melting (McCabe and Clark, 2005; Bales et al., 2006; Sirguey et al., 2009), and vegetation dynamics (Granger and Schulze, 1977; Gu et al., 2002).

Representing these processes at increasingly fine scales is
15 the goal of state-of-the-art land components of Earth system models (ESMs) (i.e., land models). However, global circulation models (GCMs) routinely compute shortwave radiative fluxes based on plane-parallel (PP) radiative transfer schemes, which do not account for the effect of topography.
This discrepancy poses a challenge for adequately capturing sub-grid-scale processes in land models.

Several models have been proposed to account for the interaction of downward solar irradiance with complex topography, accounting for slope orientation and shading effects
(Isard, 1986; Hay and McKay, 1985; Duguay, 1993) and the effect of surrounding slopes (Dozier, 1980; Dubayah et al., 1990; Dozier and Frew, 1990).

A recently developed radiation parameterization was developed to predict radiative fluxes over mountainous terrain
via multiple linear regression (Chen et al., 2006; Lee et al., 2011). This approach (henceforth termed LLH) links flux corrections over mountains to a set of grid-cell-average terrain variables, which summarize the three-dimensional nature of the land surface and are used as predictors for short-
wave fluxes. The LLH parameterization for shortwave radiation over mountains has been implemented in Global Climate Models (GCMs) and the Weather Research and Forecasting (WRF) model; it has been extensively tested over the Tibetan Plateau and the western United States (Liou et al.,
2007; Essery and Marks, 2007; Gu et al., 2012; Lee et al., 2013, 2015, 2019).

However, it is expected that sub-grid variability of these topographic effects may play a relevant role given the heterogeneous nature of surface reflectivity and topographic
features at scales smaller than the typical GCM grid cell. The importance of accounting for sub-grid-scale topography when correcting shortwave radiative fluxes over mountains was recently pointed out by an application of LLH to the DOE E3SM Exascale Earth System Model, varying model
resolution over a range of scales relevant for land processes (Hao et al., 2021).

This problem is especially relevant since in recent years the development of land models has increasingly been focused on the description of sub-grid variability of terrain
properties (Tesfa and Leung, 2017; Chaney et al., 2018). For example, in the current iteration of the Geophysical Fluid Dynamics Laboratory (GFDL) land model, this objective is achieved by summarizing grid cell heterogeneity in sets of land units (termed "tiles") characterized by approximately homogeneous physical features (Shevliakova et al., 2009; 60 Milly et al., 2014; Zhao et al., 2018; Dunne et al., 2020), including elevation, land cover, soil properties, and other environmental variables (Chaney et al., 2018). In the GFDL land model, such a sub-grid representation has not been yet tailored to describing the interaction of shortwave radi- 65 ation with topography. The objective of this work is bridging this discrepancy and developing a sub-grid parameterization for the effects of radiation over complex topography in the GFDL land model.

The sub-grid model structure is constructed using a hi- 70 erarchical multivariate clustering approach (Chaney et al., 2016, 2018) to partition land domains in a set of clusters or tiles. Tiles are here defined as land units characterized by homogeneous topographic effects with respect to downward shortwave radiation. Thus, the terrain variables used for 75 clustering land surfaces encode the physical mechanisms determining the spatial variability of radiative fluxes, such as shading and reflection from nearby slopes.

This clustering approach provides a parsimonious way to include high-resolution terrain information in global ESM 80 simulations while limiting the number of sub-grid element employed. For each terrain tile, characterized by homogeneous terrain properties, we then develop an average correction to the downward solar fluxes to account for the effects of local topography. This approach thus bridges the gap be- 85 tween the scale at which radiation and other physical processes are represented in the GFDL ESM and allows us to study how the sub-grid heterogeneity of these processes impacts the long-term evolution of the coupled physical system.

In the following, we present this new methodology and 90 test it over three mountainous sites located in different geographic regions (Alps, Andes, Himalayas), showing how model resolution and number of tiles impact the performance of the methodology. Given the wide range of spatial scale involved in the description of sub-grid topography, we eval- 95 uate the performance of the parameterization across scales, focusing in particular on the possible nonlinear dependence of incident radiation on topographic features and validating the model over independent sites.

The paper is organized as follows: we first review the ex- 100 isting parameterization for radiation over rugged terrain (Lee et al., 2011) and tailor it to our problem. This in turn requires training a model to predict topography-driven corrections for radiative fluxes based on terrain properties. Finally, we present the clustering algorithm used to divide the do- 105 main study in tiles, so as to compute local flux corrections over homogeneous regions.

The approach is then validated using different domains for independent training and testing of the methodology (Sect. 3.2), exploring the effects of terrain resolution and 110

possible consequences of nonlinear radiation–topography interactions. Finally, we explore how different tiling structures with increasing resolution improve the representation of the spatially varying radiation fields over mountains (Sect. 3.4). We close by discussing assumptions and limitations of the proposed methodology and suggesting future developments.

## 2 Methods

In order to properly account for the effects of land heterogeneity on the radiative transfer in ESMs, key variables must be obtained from high-resolution terrain datasets and properly summarized in order to capture their fine-scale effects on shortwave radiation fluxes. To this end, here we start by defining the radiative and terrain variables used to predict radiation over 3D topography. Then, we describe (i) the Monte Carlo ray-tracing algorithm used for training and testing the predictive model, (ii) the terrain-clustering algorithm used to classify land units based on the local topographic effects on radiation fluxes, and (iii) the predictive models used to link terrain properties to radiative fluxes in each land cluster. Together, these three steps provide a framework for computing fine-scale corrections to the shortwave radiation received by sub-grid land units in the GFDL land model.

### 2.1 Characterizing shortwave radiation over mountainous terrain

A parameterization explicitly accounting for the effects of 3D topography on the shortwave radiation budget was proposed by Lee et al. (2011) based on the results of Monte Carlo photon-tracing simulations (Chen et al., 2006). Here we adopt a similar approach and, following the formalism introduced by Lee et al. (2011), classify the shortwave radiation incident at a target point at the surface into five distinct components: the direct and diffuse downward solar fluxes ($F_{\mathrm{dir}}$ and $F_{\mathrm{dif}}$), and their terrain-reflected counterparts ($F_{\mathrm{rdir}}$, $F_{\mathrm{rdif}}$), which represent, respectively, direct beam or diffuse photons reaching the target site after a single reflection from neighboring terrain. Finally, a coupled flux component ($F_{\mathrm{coup}}$) consists of photons first reflected by the surface and then either back-scattered by the atmosphere or reflected multiple times by the surface before reaching the target site. For a flat surface $F_{\mathrm{rdir}}$, $F_{\mathrm{rdif}} = 0$, while $F_{\mathrm{dir}}$, $F_{\mathrm{dif}}$, $F_{\mathrm{coup}} \neq 0$ in general. We note that in the GFDL land model, diffuse radiation received by a flat surface with albedo $\alpha$ ($F_{\mathrm{dif},[\mathrm{LM}|\alpha]}$) corresponds here to the sum of $F_{\mathrm{dif}}$ and $F_{\mathrm{coup}}$. We note that these quantities can be computed separately by first computing the diffuse flux corresponding to a black surface ($F_{\mathrm{dif},[\mathrm{LM}|\alpha=0]}$), in which case the coupled flux is zero, and by then computing the coupled flux for the actual land surface as $F_{\mathrm{coup}} = F_{\mathrm{dif},[\mathrm{LM}|\alpha]} - F_{\mathrm{dif},[\mathrm{LM},\alpha=0]}$. Conversely, the diffuse flux can be obtained as $F_{\mathrm{dif}} = F_{\mathrm{dif},[\mathrm{LM}|\alpha=0]}$. Based on this formalism, the normalized flux differences between the tra-

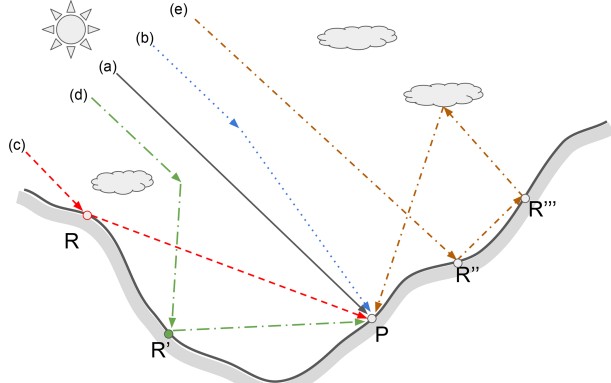

**Figure 1.** Conceptual representation of the five flux components used to characterize the nature of downward shortwave fluxes over rugged terrain, following the formalism used by Chen et al. (2006) and Lee et al. (2011). The figure includes the direct radiation flux **(a)** and diffuse radiation **(b)**, consisting of photons which are absorbed at the surface target $P$ after undergoing atmospheric scattering. Direct-reflected **(c)** and diffuse-reflected **(d)** fluxes represent photons that are reflected once at the surface. Finally, the coupled **(e)** flux component includes light undergoing multiple reflections at the surface or reflection at the surface and then atmospheric scattering. For all components, the figure shows paths incident at a point $P$ at the surface.

ditional plane-parallel (PP) case and the topography-aware case (3D) are the object of our analysis, which can be used to correct the shortwave radiative balance in ESMs. Following Lee et al. (2011), these quantities are expressed as follows.

$$
f_{\mathrm{dir}} = \frac{F_{\mathrm{dir}}^{(\mathrm{3D})} - F_{\mathrm{dir}}^{(\mathrm{PP})}}{F_{\mathrm{dir}}^{(\mathrm{PP})}}; \; f_{\mathrm{dif}} = \frac{F_{\mathrm{dif}}^{(\mathrm{3D})} - F_{\mathrm{dif}}^{(\mathrm{PP})}}{F_{\mathrm{dif}}^{(\mathrm{PP})}};
$$

$$
f_{\mathrm{rdir}} = \frac{F_{\mathrm{rdir}}^{(\mathrm{3D})}}{F_{\mathrm{dir}}^{(\mathrm{PP})}}; \; f_{\mathrm{rdif}} = \frac{F_{\mathrm{rdif}}^{(\mathrm{3D})}}{F_{\mathrm{dif}}^{(\mathrm{PP})}}; \; f_{\mathrm{coup}} = \frac{F_{\mathrm{coup}}^{(\mathrm{3D})} - F_{\mathrm{coup}}^{(\mathrm{PP})}}{F_{\mathrm{coup}}^{(\mathrm{PP})}} \quad (1)
$$

Here the direct-reflected and diffuse-reflected components are normalized with respect to the corresponding non-reflected flux component since they are equal to zero in the plane-parallel case (Lee et al., 2011). These quantities are defined in Eq. (1) for a given surface albedo value. While direct and diffuse component are independent of surface albedo, the reflected flux components are linearly dependent on albedo. Finally, the coupled flux is nonlinearly dependent on surface albedo. A schematic representation of these flux components is reported in Fig. 1. Predicting these five $f_i$ terms over land tiles representing heterogeneous terrain properties is the objective here. To this end, a predictive model linking the $f_i$ terms to tile terrain properties is necessary. To train such a model, we use ray-tracing simulations, which are discussed next.

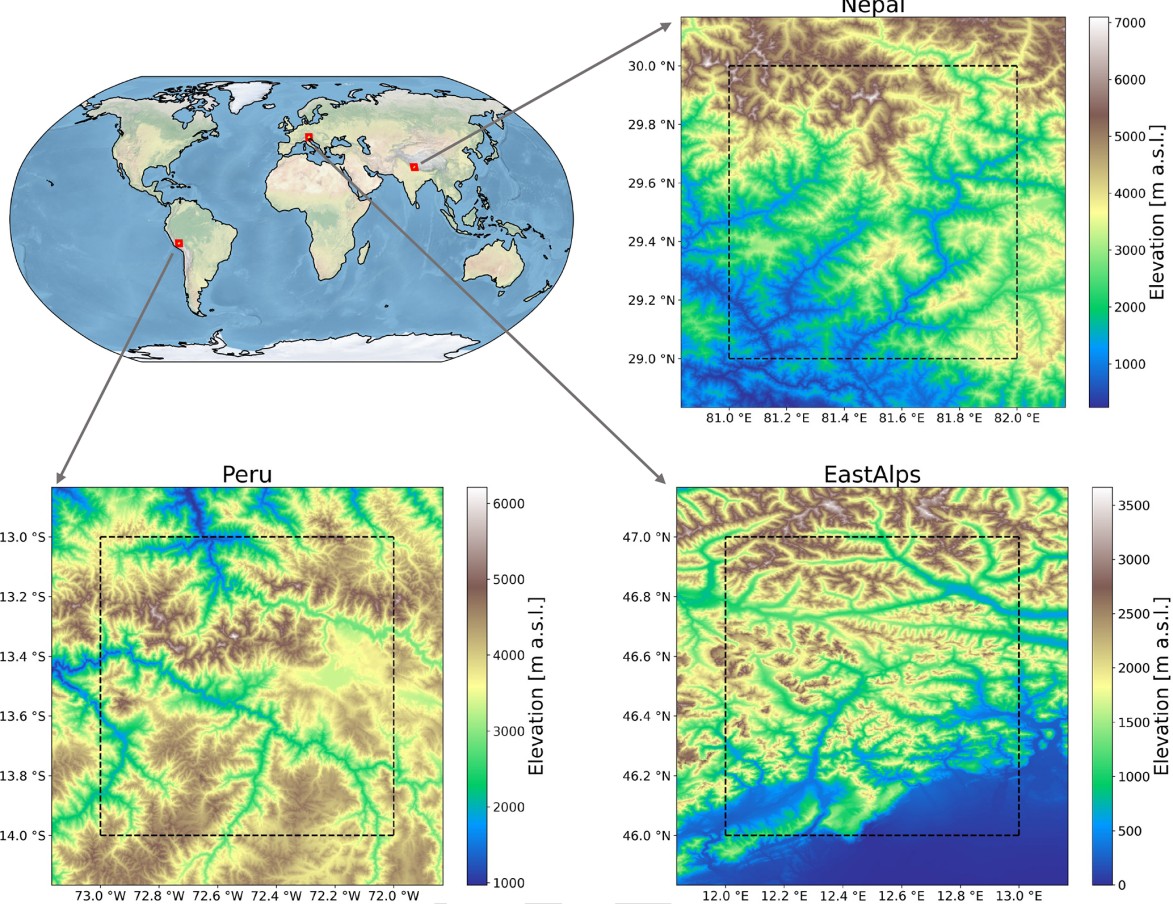

**Figure 2.** Location of the three land domains selected for this study. For each location, the 90 m resolution digital elevation used in the study is shown.

## 2.2   A Monte Carlo ray-tracing algorithm

Due to the complex interactions involved, topographic effects on shortwave radiation are generally studied based on Monte Carlo (MC) ray-tracing techniques, which approximate the three-dimensional radiative transfer process by tracking the fate of a large number of photons (Miesch et al., 1999; Chen et al., 2006; Mayer et al., 2010).

Here, in order to develop a predictive model for the 3D radiation correction terms, we employed a MC algorithm (sometimes referred to as photon-tracing or ray-tracing algorithm) to approximate the true physics of radiation–topography interactions. These high-resolution simulations are used to calibrate the predictive models for topographic effects over each tile and to validate the proposed parameterization. The MC scheme explicitly describes the interaction of downwelling shortwave radiation with a 3D surface corresponding to a region characterized by complex topography. The algorithm was implemented in a software package, which is made available online (Zorzetto, 2022a). The MC method has been widely used to study radiation interaction with 3D surfaces (Chen et al., 2006; Lee et al., 2011; Mayer,

2009; Mayer et al., 2010; Villefranque et al., 2019; Larue et al., 2020). The MC model adopted here broadly follows previous models developed by Chen et al. (2006) and Mayer (2009).

In our MC algorithm, photons are randomly released at the top of atmosphere (TOA) and travel in a direction determined by the Sun's zenith $\theta_0$ and azimuth $\phi_0$ angles. After a path of random length, which depends on the optical properties of the medium, the photon encounters scattering or absorption based on the single-scattering albedo properties of the atmospheric constituents (Fu and Liou, 1992; Liou, 2002). In the present simulation, we used for each site optical properties computed from the RRTMGP radiation code (Pincus et al., 2019) using the GFDL AM4 model (Zhao et al., 2018). We note that the Monte Carlo model is run offline, prescribing the atmospheric profile and solar position in each model run. The atmospheric column used in the model is composed of 34 layers characterized by optical properties, which encode the absorption and Rayleigh scattering of photons by gas molecules. We limit our analysis to aerosol-free and clear-sky conditions, similar to previous studies (Chen et al., 2006;

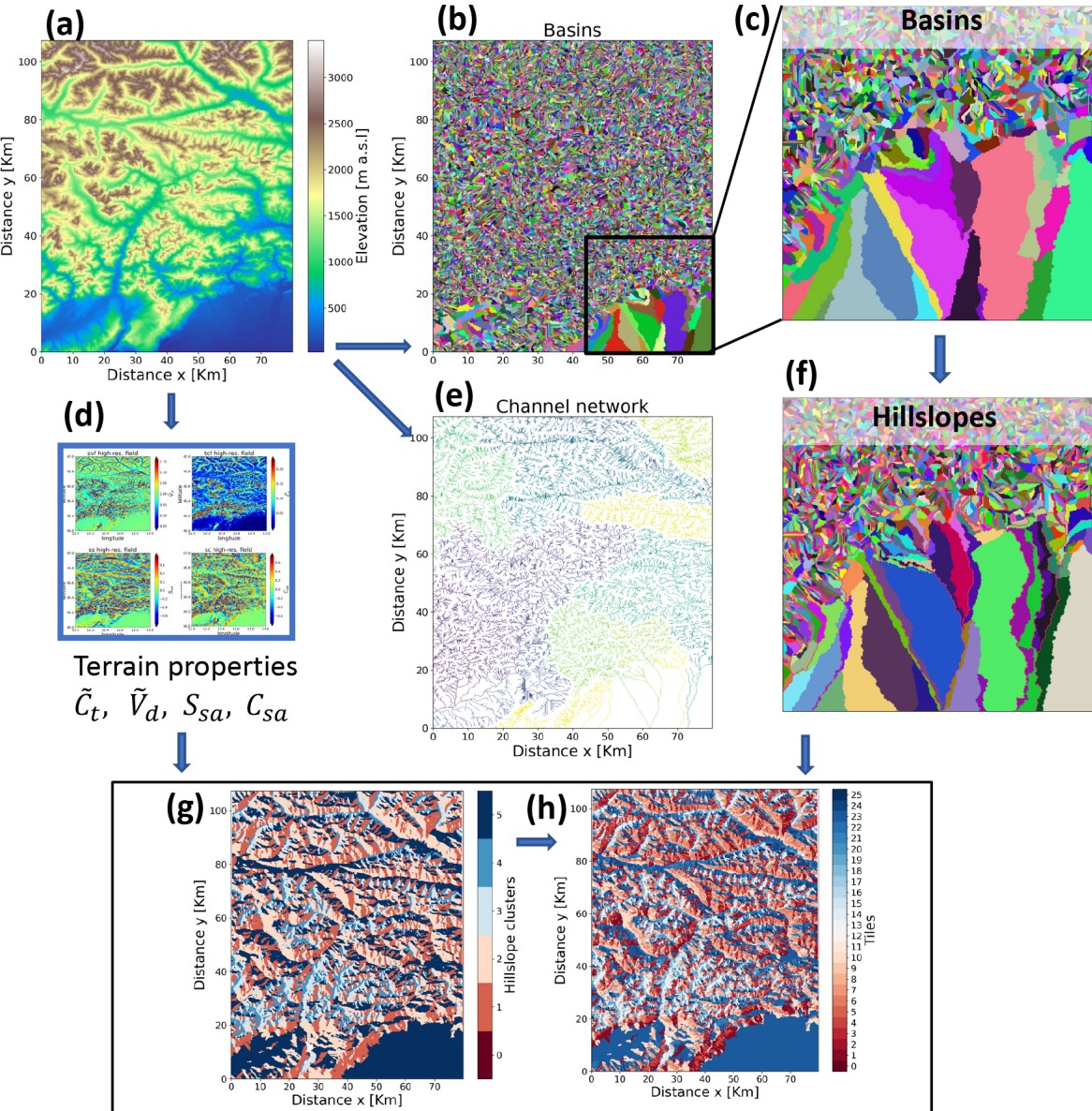

**Figure 3.** Schematic representation of the land clustering workflow. The land elevation map **(a)** is used to compute the variables of interest (sky and terrain view factors and functions of terrain slope and aspect, **d**) which are then used in the hierarchical clustering steps, yielding a map of homogeneous land units needed to parameterize radiation–topography interactions. The delineation of channel networks **(e)**, watersheds **(b, c)**, and hillslopes **(f)** follows the approach developed by (Chaney et al., 2018). The resulting spatial distribution of hillslope clusters and land tiles (corresponding to the two stages of land clustering) are shown in **(g)** and **(h)**, respectively.

Lee et al., 2011). Therefore, atmospheric properties are completely determined by optical depths and single-scattering albedo at each level (Liou, 2002). The lower boundary of the simulation domain is derived from high-resolution (90 m) terrain model derived from the Shuttle Radar Topography Mission (SRTM) dataset (Farr et al., 2007). SRTM elevation fields are used to construct a three-dimensional surface characterized by triangular mesh elements characterized by uniform albedo and Lambertian reflection. In the following we refer to "pixel" as the image elements of the high-resolution

input digital elevation maps, to clarify the difference with land model grid cell and land model sub-grid units, termed "tiles" in the following.

In each MC simulation, photons are tracked from the TOA until they are absorbed or leave the simulation domain. For each photon, interactions with atmospheric constituents (scattering or absorption) and with land surface elements (absorption or reflection) are used to characterize the nature of radiation incident over surface elements. Tracking the path of each photon, the downward irradiance is then decomposed

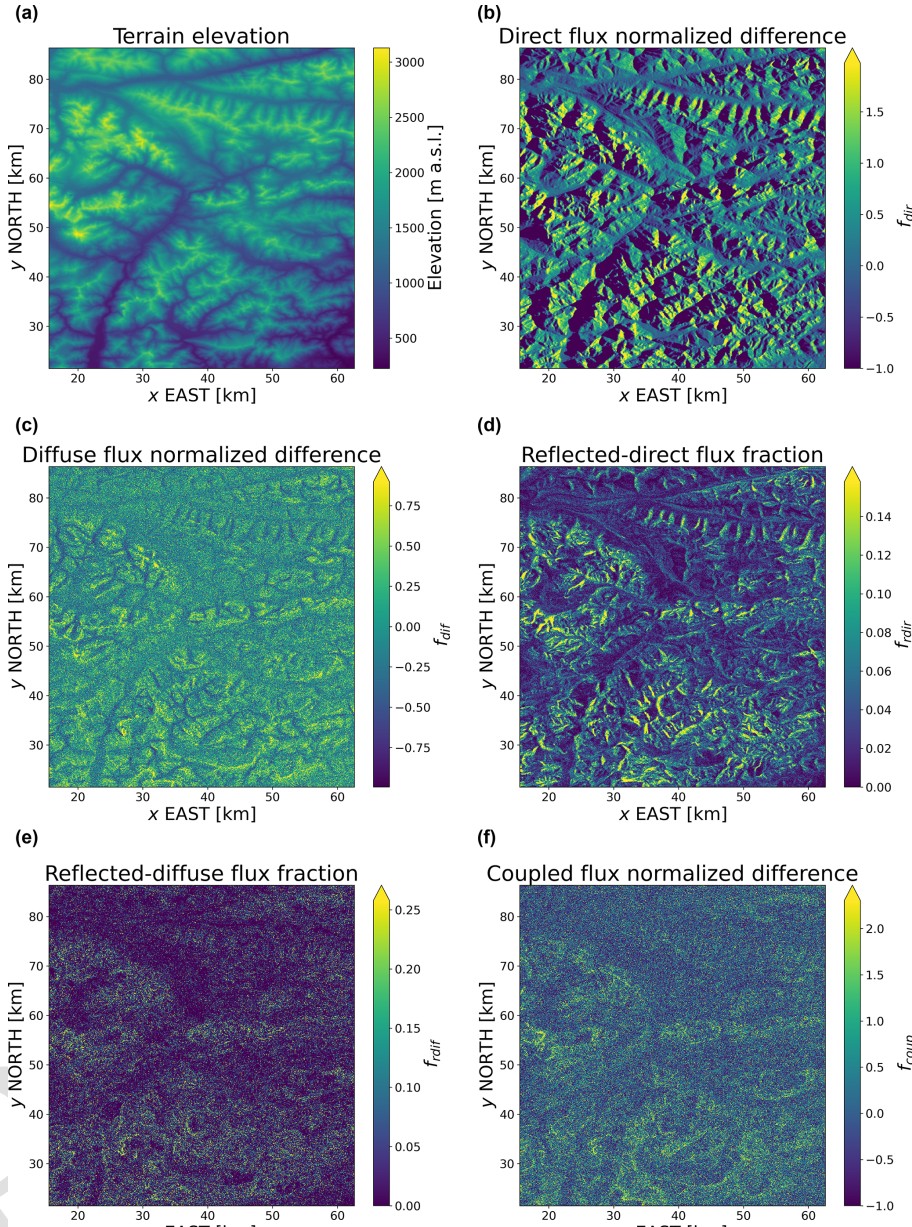

**Figure 4.** Normalized differences in downward fluxes obtained running Monte Carlo simulations using the real topography and a flat domain, respectively ($\mu_0 = 0.40$, $\phi_0 = \pi/2$). Maps are shown using an equal-area Mollweide projection.

into the five flux terms introduced in Sect. 2.1. If $E_0$ is the radiation incident at the TOA with a cosine of the zenith angle $\mu_0 = \cos\theta_0$, then the horizontal distribution of solar irradiance received by the land surface is given by (Mayer, 2009)

$$E_{k,l} = E_0 \cos\theta_0 \frac{1}{N} \frac{A}{A_{k,l}} \sum_{i=1}^{N_{k,l}^{(s)}} w_i, \tag{2}$$

where $w_i$ is the energy of the $i$th incident photon, $N$ is the total number of photons tracked in the simulation over a do-main with area $A$, and $N_{k,l}^{(s)}$ is the number of photons absorbed by the surface within grid cell $(k,l)$ with area horizontal plane surface area $A_{k,l}$. Photons are released with unit energy at the TOA and lose a fraction of this energy through absorption in each atmospheric layer (Mayer, 2009) or through absorption at the ground. Therefore, Eq. (2) evaluates the spatial distribution of solar irradiance based on the density of photons that are absorbed at the surface. While a 3D mesh is used in the Monte Carlo simulations, the area of the cell $A_{(k,l)}$ is defined as the area of the cell on the horizontal plane. Equation (2) expresses the radiation received by a

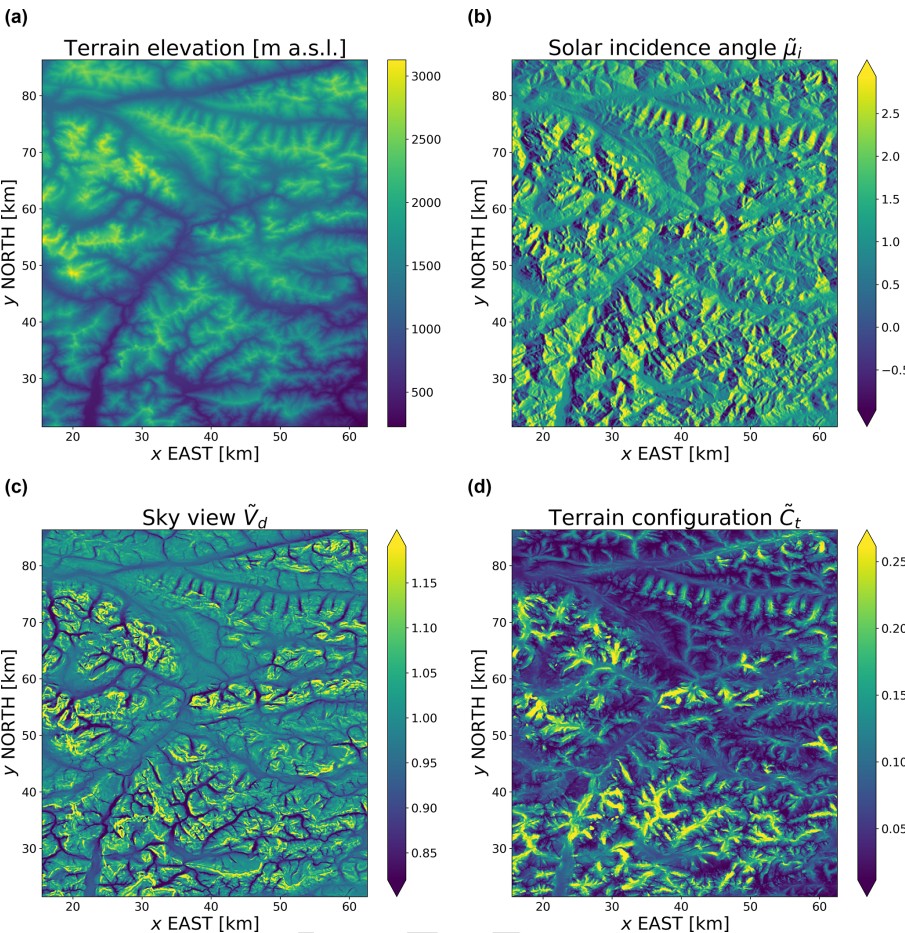

**Figure 5.** Terrain variables computed for the same domain shown in the previous figure (eastern Alps). Solar incidence angle was computed for $\mu_0 = 0.40$, $\phi_0 = \pi/2$. Maps are shown using an equal-area Mollweide projection.

single land surface cell as a fraction of the radiation flux at the top of the surface $E_0$ by summing the energy $w_i$ of all the photons absorbed by the surface over that area. Since the interactions of each photon are tagged (e.g., keeping track of atmospheric scattering and any previous reflections at the surface) the radiation received can be directly classified in one of the five flux components, as defined in Eq. (1).

For two independent domains located in the Alps and Peru (Fig. 2), the MC calculations were repeated for six solar zenith angle values ($\cos\theta_0 \in \{0.1, 0.25, 0.4, 0.55, 0.7, 0.85\}$), four solar azimuth angles ($\phi_0 \in \{0, \pi/2, \pi, 3\pi/2\}$), and a uniform surface reflectivity value set to $\alpha_s = 0.3$. While only two domains are used for MC calculations due to the computational expense of this procedure, an additional domain located in high mountain Asia (Fig. 2) is used to further test the results of the spatial clustering over areas with different topographic features.

## 2.3 Predicting radiative fluxes over complex terrain

Over mountainous regions, the local irradiance at the surface can exhibit significant departures from its areal-average value at spatial scales routinely resolved in ESMs due to the complexity of topography and surface properties. In order to develop a simple parameterization to explain the magnitudes of these departures over mountains areas, we need predictor variables encoding the interaction between downward radiation and topographic features. For this purpose, here we define a set of relevant variables following previous work by Chen et al. (2006) and Lee et al. (2011). The terrain variables used to predict downward fluxes are (i) the sky view factor $V_d$, which represents the ratio of diffuse sky irradiance at a point on an unobstructed horizontal surface under the assumption of isotropic diffuse radiation (Dozier and Frew, 1990; Helbig et al., 2009); (ii) the terrain configuration $C_t$, which quantifies the contribution to the irradiance at a point originated by reflections from surrounding mutually visible slopes; and (iii) the solar incident angle $\mu_i$, which is the angle between the direct solar beam and the normal to the

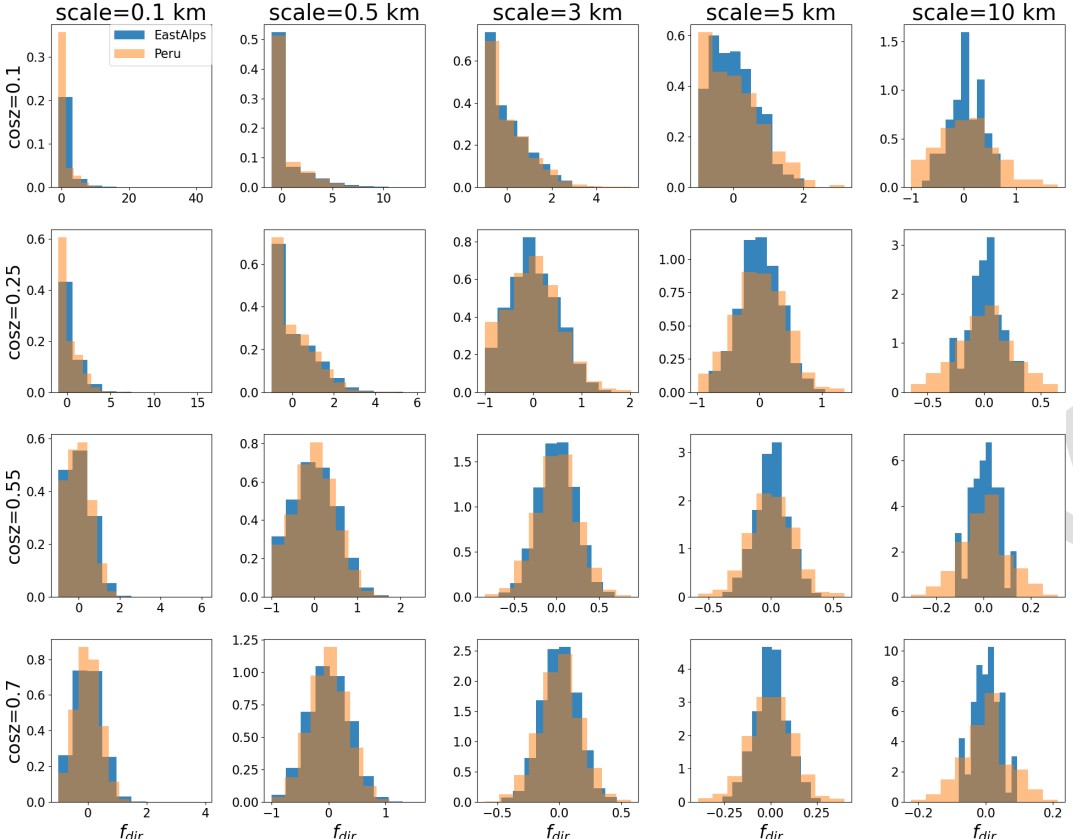

**Figure 6.** Histograms of the direct radiation correction $f_{\text{dir}}$ between the 3D and PP cases at varying zenith angles and spatial averaging scales for the Peru and eastern Alps domains.

surface. These terrain variables are derived from the Shuttle Radar Topography Mission (SRTM, Farr et al., 2007) high-resolution (90 m) terrain information.

In order to compute the sky view factor, we use the rapid procedure proposed by Dozier and Frew (1990), whereby the unobstructed fraction of sky hemisphere is approximated as

$$
V_{\text{d}} \simeq \frac{1}{2\pi} \int_{0}^{2\pi} \Big[ \cos\theta_{\text{s}} \sin^2 H_{\phi} + \sin\theta_{\text{s}} \cos(\phi - \phi_{\text{s}})
$$

$$
\big( H_{\phi} - \sin H_{\phi} \cos H_{\phi} \big) \Big] d\phi, \tag{3}
$$

for a point with slope $\theta_{\text{s}}$, aspect $\phi_{\text{s}}$, and horizon angle $H_{\phi}$ (i.e., angular distance between zenith and local horizon along the generic azimuth direction $\phi$). The terrain configuration, which quantifies the reflected radiation received by surrounding slopes in direct sight, can then be obtained as $C_{\text{t}} \simeq (1 + \cos\theta_{\text{s}})/2 - V_{\text{d}}$ following the approach by (Dozier and Frew, 1990).

While fields of $V_{\text{d}}$ and $C_{\text{t}}$ can be computed offline for any given elevation map, the Sun's incidence angle on a surface does depend on the Sun's position through the local zenith

and azimuth angles $(\theta_0, \phi_0)$

$$
\mu_i / \cos\theta_{\text{s}} = \cos\theta_0 + \sin\theta_{\text{s}} \tan\theta_0 \cos(\phi_{\text{s}} - \phi_0)
$$

$$
= \mu_0 + \sin\theta_0 \left( S_{sa} \sin\phi_0 + C_{sa} \cos\phi_0 \right), \tag{4}
$$

where topographic information is encoded in the two terms $S_{sa} = \sin\theta_{\text{s}} \sin\phi_{\text{s}}/\cos\theta_{\text{s}}$ and $C_{sa} = \sin\theta_{\text{s}} \cos\phi_{\text{s}}/\cos\theta_{\text{s}}$. Here $\mu_i$ is the cosine of the solar incidence angle (i.e., angle between the incoming direct light beam and the normal to the land surface), while $\mu_0$ is the cosine of the solar zenith angle (i.e., the incidence angle with respect to a horizontal plane). Additionally, for parameterizing the effect of topography on diffuse radiation, we use a standardized elevation $h_n = (h - \mu_h)/\sigma_h$ obtained by normalizing elevation based on its grid cell average $\mu_h$ and standard deviation $\sigma_h$.

For the purpose of parameterizing solar fluxes, we divide these terrain variables by the local terrain slope obtaining the normalized variables $\tilde{\mu}_i = \mu_i/\cos\theta_{\text{s}}$, $\tilde{V}_{\text{d}} = V_{\text{d}}/\cos\theta_{\text{s}}$, and $\tilde{C}_{\text{t}} = C_{\text{t}}/\cos\theta_{\text{s}}$ as recommended by (Lee et al., 2013). Note that while $V_{\text{d}} \in [0, 1]$, the ratio $\tilde{V}_{\text{d}} = V_{\text{d}}/\cos\theta_{\text{s}}$ can sometimes be larger than 1. In previous work (Lee et al., 2011) these terrain predictors are averaged over an area representative of an entire model grid cell and used as predictors to derive average correction terms $f_i$ for the five flux components

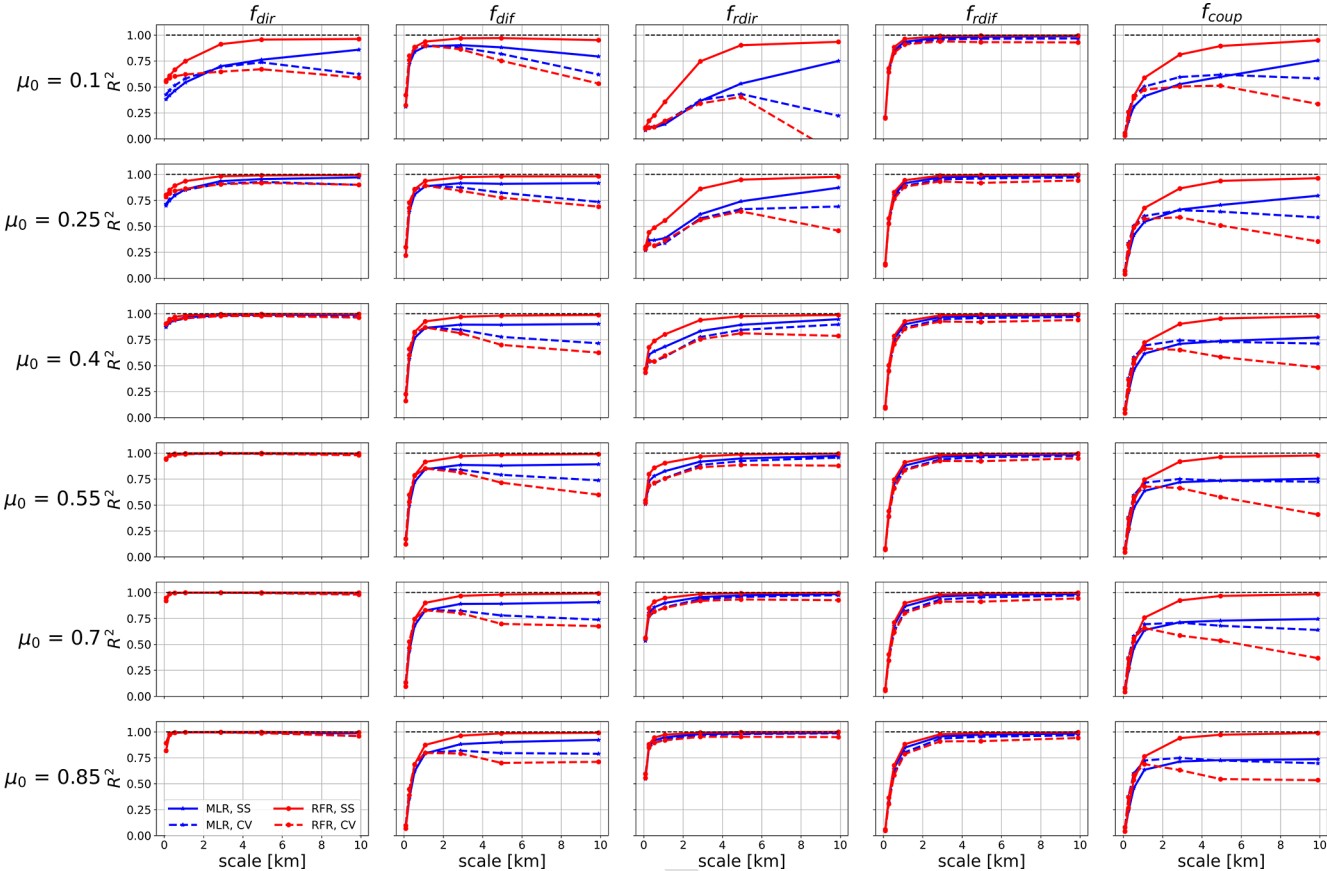

**Figure 7.** Coefficient of determination $R^2$ for the two different predictive models: multiple linear regression (MLR, blue lines) and random forest regression (RFR, red lines). Results are reported for varying averaging scales and cosines of the solar zenith angle ($\mu_0$). The models were trained over the Peru site and tested over the same site (SS, continuous lines) and over the independent eastern Alps site for cross validation (CV, dashed lines).

introduced in Sect. 2.1. Here, high-resolution fields of these predictor variables are first computed based on the original 90 m digital elevation maps. These fields are then employed to inform the partitioning of the land domain into a set of tiles for which we expect topography to have a similar effect on radiation. This step is achieved using a hierarchical clustering methodology described next. Based on this partitioning, relations to predict the topographic effects on radiation (the $f_i$'s) will be applied to each tile independently.

## 2.4 Hierarchical clustering of terrain properties

In order to capture the spatial variability of radiative fluxes, here we employ a hierarchical multivariate clustering approach (HMC) which was recently introduced to study the role of heterogeneity in hydrological and land models (Chaney et al., 2016, 2018). Here we tailor HMC to the case of shortwave radiative fluxes by performing the land clustering based on terrain properties (namely $\tilde{\mu}_i$, $\tilde{V}_d$, and $\tilde{C}_t$), which are known to modulate the downwelling radiation over mountains as discussed in the previous section.

The land fraction of the study sites, which are chosen to represent an ESM grid cell that is typical in size, is first divided into a maximum of three components: soil, glacier, and lake. The soil fraction is then subdivided into a set of tiles characterized by homogeneous terrain properties relevant for capturing the effects of topography on radiative transfer. Additionally, lake and glacier areas, where present, are treated as individual separate tiles. We note that in the domains selected for this study, lake and glacier areas constitute a small fraction of the total grid cell area. When applying the methodology to areas where glaciers cover a large fraction of the grid cell, it may be useful to also partition glacier areas into multiple clusters. This can be done following the same methodology described here, since for the purpose of radiation–terrain interactions the only relevant parameter would be the average albedo of each cluster (land or glacier). In our approach, the land clustering is based on four terrain variables: normalized sky view factor, terrain configuration, $S_{sa}$, and $C_{sa}$. Note that these variables are independent of the Sun's position. Once the direction of the incoming beam is given ($\phi_0$ and $\cos\theta_0$), average values of $S_{sa}$ and $C_{sa}$

over any given tile uniquely identify the solar incident angle for each point on the land surface by means of Eq. (4).

A conceptual summary of this clustering procedure is described in Fig. 3. The digital elevation map (Fig. 3a) is used to compute the drainage network (Fig. 3e) necessary to partition the domain in basins (Fig. 3b, c) based on a threshold area of $1 \times 10^5 \mathrm{m}^2$. Basins are in turn subdivided into hillslopes (Fig. 3f) following Chaney et al. (2018): each basin is divided into up to three contiguous hillslope elements, corresponding to the left side, right side, and headwaters.

Then, hillslope elements are aggregated into $k$ "characteristic hillslopes" via k-means clustering (MacQueen et al., 1967) in the four-dimensional space of the variables $\tilde{V}_\mathrm{d}$, $\tilde{C}_\mathrm{t}$, $S_{sa}$, and $C_{sa}$. This enables us to obtain land units characterized by similar radiation–topography interactions. Figure 3g shows the spatial distribution of these characteristic hillslope clusters for the case $k = 5$.

Finally, each of these $k$ land units is further partitioned into $p$ sub-units by a second application of the k-means clustering algorithm based on the four variables $\tilde{V}_\mathrm{d}$, $\tilde{C}_\mathrm{t}$, $S_{sa}$, and $C_{sa}$. In Fig. 3h we show the result of this procedure, which yields 25 tiles in this example.

In this application any areas covered by glaciers or lakes are treated as separate land units. Therefore, in the current configuration the number of tiles ($n_\mathrm{t}$) used to describe land heterogeneity within a single land model grid cell varies between $k \cdot p$ and $k \cdot p + 2$ depending on the presence of lake and glacier units. The reason for the first step (subdivision of land in characteristic hillslopes) originates from the desire of partitioning land in hydrologically coherent units. In the context of shortwave radiation received by land, it is important to understand what the effects and the benefits of this multi-level clustering are. Therefore, we will test the methodology for different tile configurations over the same domains.

A natural test for the ability of the tiled grid to reproduce the actual spatial distribution of solar radiation can be performed as follows. We here test the results for multiple HMC configurations obtained by varying the number of characteristic hillslopes ($k$), as well as the number of land units within each characteristic hillslope ($p$). For illustration purposes, we consider the following two cases: a fixed value of $k = 5$ and a varying $p$ and the opposite (varying $k$, setting $p = 5$). This experiment leads to a set of grid configurations with a number of tiles per grid cell varying from 5 to 1000, with different weights given to the first level (partitioning of land in hillslopes) and the second level, in which each characteristic hillslope is further subdivided into $p$ homogeneous land units contributing to the overall number of tiles $n_\mathrm{t}$. This experiment thus elucidates the relative performance of the two different levels of the hierarchical clustering approach in capturing the spatial heterogeneity of the domain.

Therefore, based on this procedure, the land model grid cells are subdivided into a number of tiles $n_\mathrm{t}$. A maximum of two additional tiles can be included representing lake and a glacier areas if these are present in a given grid cell. Glacier

boundaries are determined using the GLIMS database (Raup et al., 2007). Each tile is characterized by statistically homogeneous values of the variables of interest for 3D radiative transfer. Over each tile $t = 1, \ldots, n_\mathrm{t}$, we then compute the average value of each predictor variable. For a generic variable $\Theta \in \left\{ \tilde{V}_\mathrm{d}, \tilde{C}_\mathrm{t}, S_{sa}, C_{sa} \right\}$ we have

$$\langle \Theta \rangle_\mathrm{t} = \frac{\sum_{i=1}^{n_\mathrm{p}} A_i \Theta_i 1_{\mathrm{t},i}}{\sum_{i=1}^{n_\mathrm{p}} A_i 1_{\mathrm{t},i}}, \tag{5}$$

with the $< \cdot >_\mathrm{t}$ operator representing the operation of average over the points classified as part of tile t, $1_\mathrm{t}$ is the indicator function selecting tile t ($1_{\mathrm{t},i} = 1$ if pixel $i$ of the high-resolution terrain map belongs to tile t, $1_{\mathrm{t},i} = 0$ otherwise). $A_i$ and $\Theta_i$ are the area and value of the property $\Theta$ computed for the $i$th pixel in a high-resolution terrain map with $n_\mathrm{p}$ pixels. For the solar incidence angle, once the solar position is known we can write its average value over a tile t as

$$\langle \tilde{\mu}_i \rangle_\mathrm{t} = \mu_0 + \sin\theta_0 \left( \langle S_{sa} \rangle_\mathrm{t} \sin\phi_0 + \langle C_{sa} \rangle_\mathrm{t} \cos\phi_0 \right). \tag{6}$$

Therefore, the average values of these quantities over each tile represent the "characteristic" value of the properties over that tile and can be used to summarize the effect of multi-scale radiative transfer over mountainous regions.

## 2.5 A predictive model for the flux correction terms

The final step needed in order to parameterize the effect of complex topography on incident shortwave radiation is specifying a predictive model to link the terrain variables defined in Sect. 2.3 to the five flux components (Sect. 2.1). This step will enable us to extend the results of costly ray-tracing simulations at the global scale and provide predictive equations that can be directly applied to high-resolution terrain maps as well as to the local averages from the tiling structure defined in Sect. 2.4.

Here we focus on two different approaches, namely a multiple linear regression model (MLR, based on the previous LLH parameterization) and an alternative approach based on random forest regression (RFR). We explore these two distinct families of statistical models in order to evaluate the potential role on nonlinearity for different radiation flux terms, keeping in mind that MLR, the simpler statistical model, is in general to be preferred due to the greater ease of interpretation it provides and its reduced computational expense when employed in Earth system models.

MC photon-tracing simulations are performed at the native 90 m resolution of terrain products available. In the following analysis, results are coarsened at a range of spatial scales up to 10 km in order to test the robustness of the proposed parameterization to the spatial scale of the terrain variable used as predictors, and explore potential dependencies of radiation–topography interaction on the spatial scale considered. In each case, both MC simulation results and terrain properties are averaged at the same spatial resolution in

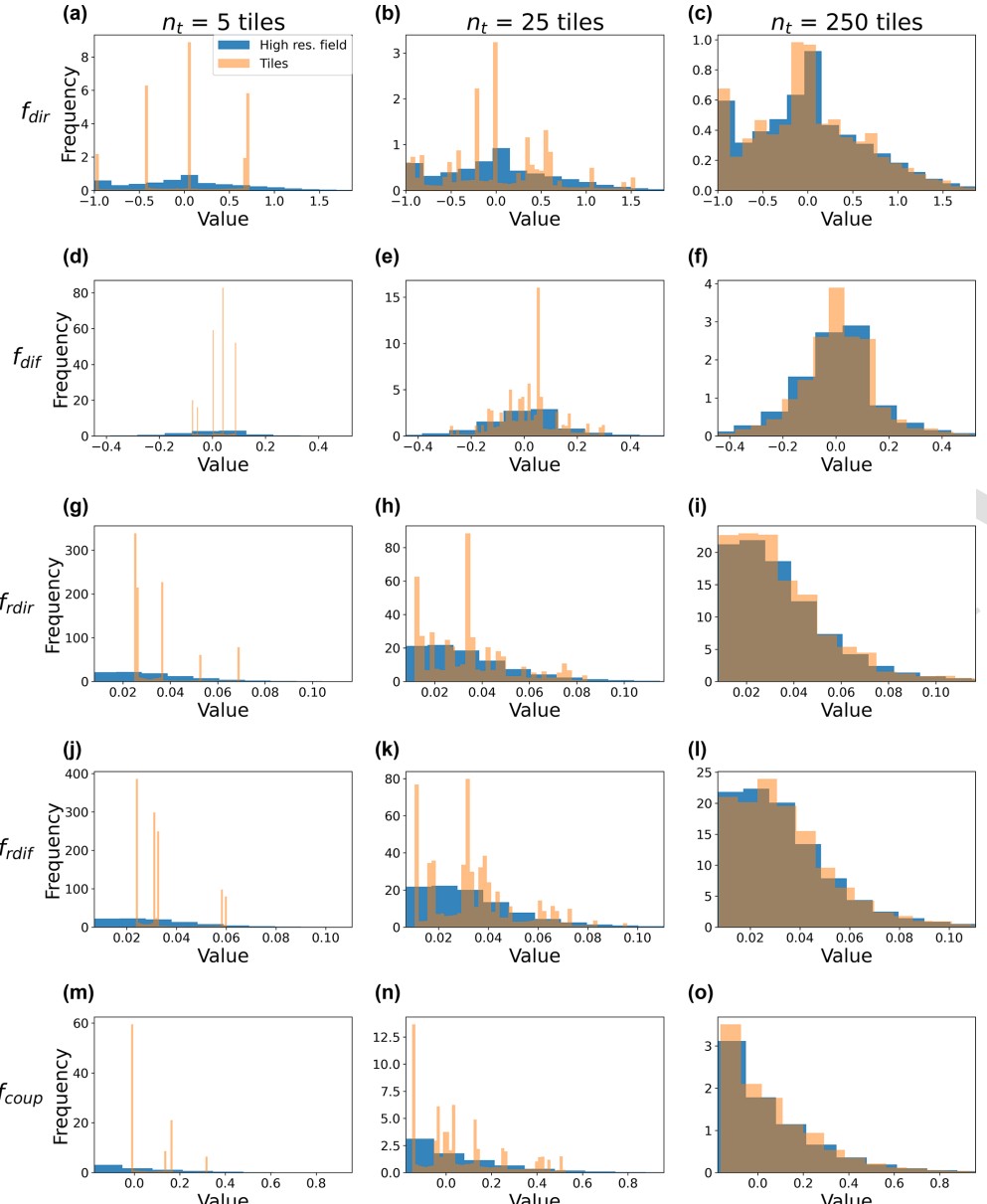

**Figure 8.** Histograms of the distribution of 3D–PP incident radiation differences for varying number of tiles. Average 3D correction over each tile (orange) compared with the frequency distribution of the high-resolution field (blue histograms). Results are shown for the eastern Alps domain, for a given solar angle ($\mu_0 = 0.4$).

order to train and test the predictive model at that specific spatial scale. MC simulations corresponding to the 3D case (i.e., over a three-dimensional surface) and over a flat surface (plane-parallel case, PP) were subtracted in order to compare the "true" flux deviations as defined in Eq. (1). Additionally, given the finite extension of the domain used in the MC simulations, spurious effects may be present due to the periodic boundary conditions imposed, especially for the lower values of solar zenith angle. To mitigate this issue, only the central part of each simulation domain (of size $\sim 1° \times 1°$) is used to train the predictive model. A fraction of size 0.2 times the

linear dimension of the domain in discarded at each boundary.

The MC simulations were repeated for varying zenith angles (from $\mu_0 = 0.1$ to $\mu_0 = 0.85$) and four azimuth angles. The results for each azimuth are pulled together so that a single predictive model for each zenith angle is derived. For application to ESMs, the results should be interpolated when predictions for a generic zenith angle are needed.

After training each model for each solar zenith angle and flux component, we obtain a set of predictive equations link-

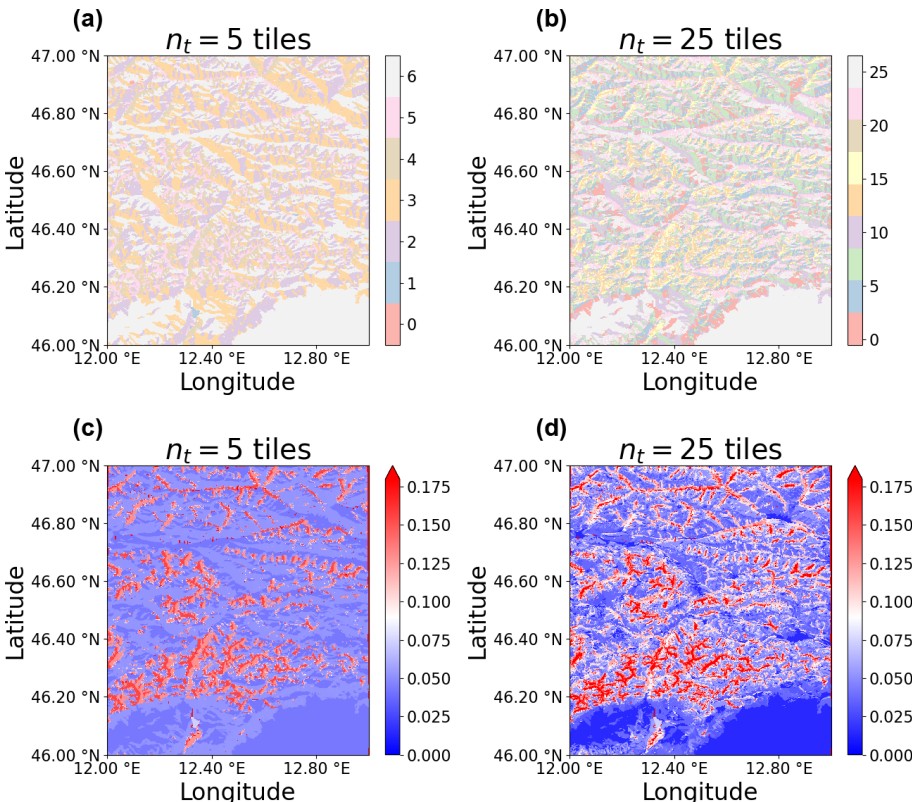

**Figure 9.** Effect of a different number of tiles for modeling the sub-grid variability of terrain properties over the study site. **(a, b)** Spatial distribution of the tiles representing homogeneous land units. Note that for this domain, in addition to the $n_t$ land clusters, there are two additional tiles corresponding to glaciers (tile 0) and lakes (tile 1), so that the resulting number of land units is 7 and 27 for **(a)** and **(b)**, respectively. **(c, d)** Spatial distribution of the terrain view factor ($\tilde{C}_t$) locally averaged over each tile.

ing the $f_i$'s (i.e., the normalized deviation between 3D and PP case) to the terrain predictors at the same spatial scale:

$$f_{\mathrm{dir}} = g_{\mathrm{dir,m,\cos\theta_0,\Delta x}}\left(\langle \tilde{V}_d \rangle_{\Delta x}, \langle \tilde{\mu}_i \rangle_{\Delta x}\right), \tag{7}$$

$$f_{\mathrm{dif}} = g_{\mathrm{dif,m,\cos\theta_0,\Delta x}}\left(\langle \tilde{V}_d \rangle_{\Delta x}, \langle \tilde{\mu}_i \rangle_{\Delta x}, \langle h_n \rangle_{\Delta x}\right), \tag{8}$$

$$_5 \quad f_{\mathrm{rdir}} = g_{\mathrm{rdir,m,\cos\theta_0,\Delta x}}\left(\langle \tilde{V}_d \rangle_{\Delta x}, \langle \tilde{C}_t \rangle_{\Delta x}, \langle \tilde{\mu}_i \rangle_{\Delta x}\right), \tag{9}$$

$$f_{\mathrm{rdif}} = g_{\mathrm{rdif,m,\cos\theta_0,\Delta x}}\left(\langle \tilde{V}_d \rangle_{\Delta x}, \langle \tilde{C}_t \rangle_{\Delta x}\right), \tag{10}$$

$$f_{\mathrm{coup}} = g_{\mathrm{coup,m,\cos\theta_0,\Delta x}}\left(\langle \tilde{V}_d \rangle_{\Delta x}, \langle \tilde{C}_t \rangle_{\Delta x}, \langle \tilde{\mu}_i \rangle_{\Delta x}\right), \tag{11}$$

where $g$ represents the parametric relations linking flux correction terms to terrain variables for different predictive mod-
10 els ($m = $ MLR or $m = $ RFR). These equations are derived for each flux component, solar zenith angle $\cos\theta_0$, and spatial scale $\Delta x$. We note that these equations were derived for a single surface reflectivity value. The direct and diffuse flux components are independent of albedo. Reflected
15 fluxes are linearly dependent on albedo, meaning that the predicted value can be rescaled by the surface albedo. Finally, $g_{\mathrm{coup}}$ is nonlinearly dependent on the surface albedo, meaning that predictions for different albedo values can, e.g., be

obtained by interpolation (Lee et al., 2011). Once the $f_i$ dimensionless predictions are available, the dimensional value 20 of shortwave fluxes over rugged terrain can finally be obtained from Eq. (1). In general, simply applying a correction to the downward radiation received by land will not ensure energy conservation. This is expected in general, as some of the 3D topographic effects parameterized here would in gen- 25 eral lead to energy fluxes between neighboring land model grid cells. A procedure was proposed by (Lee et al., 2015) that can be used to address this issue. In this approach, an effective albedo $\alpha_{3D}$ is computed for each land grid cell, such that a grid cell characterized by this $\alpha_{3D}$ and forced by 30 plane-parallel radiation (PP) absorbs the same amount of radiation as in the case of a surface characterized by actual land albedo $\alpha$, while forced by the 3D-corrected downward radiation fluxes. By returning the 3D albedo (which effectively represents the reflectivity of a "rough" but flat land surface) 35 to the atmosphere, energy is conserved while accounting for the 3D topographic correction.

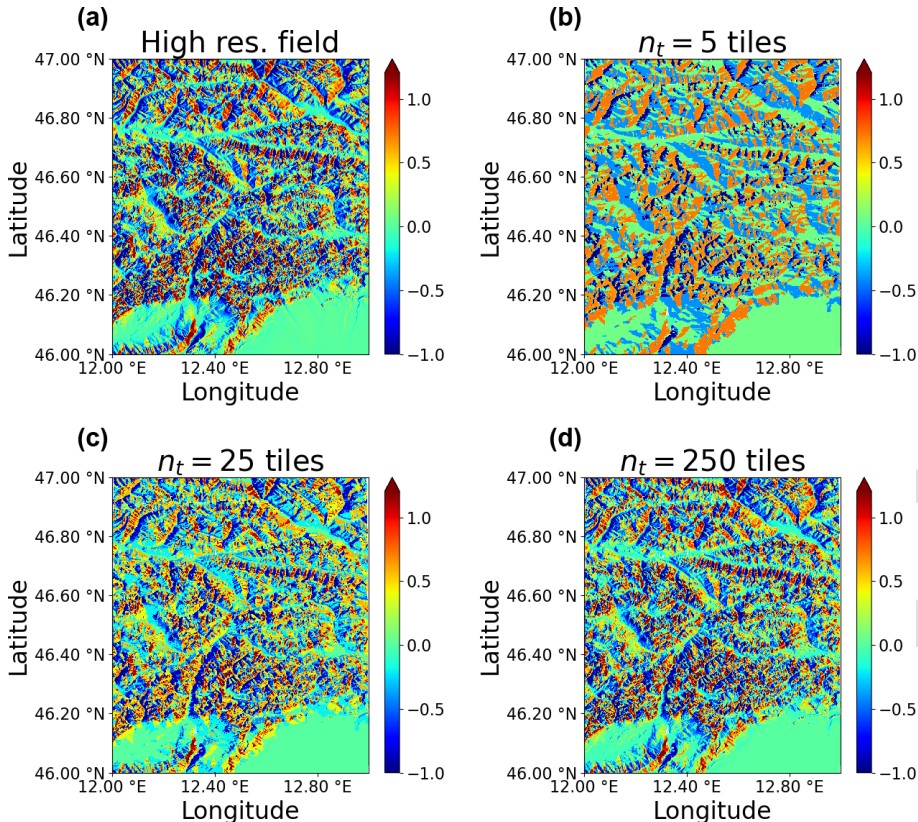

**Figure 10.** Normalized differences between 3D and PP direct fluxes for high-resolution predictions and predictions for different numbers of tiles ($k = 5, 25, 250$), obtained setting a fixed number of hillslopes and varying the number of units for each hillslope ($k = 5$). CE1

## 3 Results and discussion

### 3.1 Spatial distribution of solar irradiance

A representative output obtained from Monte Carlo ray-tracing simulations for the eastern Alps (EastAlps in Fig. 2) domain is portrayed in Fig. 4, featuring the spatial distribution of differences between 3D topography and the PP case for a given incoming solar beam direction, while fields of the corresponding terrain variables for the same domain are reported in Fig. 5.

The direct flux appears to be prominently modulated by the presence of topography, with the distribution of shaded slopes following that of solar incident angles computed based on the solar angles for the current simulation (Fig. 5). In the case of the diffuse flux, differences between the 3D and PP simulations are less apparent, while a similar behavior is observed for the reflected flux components.

From visual inspection of these simulation results, it is apparent that the spatial variation in the direct flux is primarily controlled by the solar incident angle. The diffuse flux, on the other hand, shows a spatial variability primarily consistent with that of the sky view factor, as expected based on previous studies (Chen et al., 2006). Reflected and coupled flux spatial variability appear to be more complex, potentially controlled by the terrain configuration. This exploratory analysis reveals that, with the sole exception of the direct flux component, variations in downward fluxes are arguably not explained by a single linear relation but can in general involve multiple terrain predictors and potentially nonlinear effects. This hypothesis will be tested in the next section by examining the skill of a linear and nonlinear model when describing these relationships.

For the direct flux, histograms of the normalized differences between PP and 3D simulations are featured in Fig. 6, where the simulation results have been aggregated at a range of increasing spatial averaging scales. For low solar zenith angles, the frequency distribution of $f_{\mathrm{dir}}$ tends to have an atom at $-1$, a lower boundary value corresponding to the case of complete shade. This is clearly a limitation for the linear model approach, since this behavior imposes a nonlinear relation between $f_{\mathrm{dir}}$ and terrain variables (solar incident angle and sky view). However, averaging the results at increasing spatial scales we see this effect progressively diminishes, as the probability of a complete shade decreases. For both the domains examined in Fig. 6, this behavior is similar and similarly decreases with the cosine of the solar zenith angle. We note that at spatial scales larger than approximately 5 km the effect disappears. We note that it is a

this range of scales that previous parameterization of 3D radiation over mountains were trained (Lee et al., 2011).

## 3.2 Model evaluation: sensitivity to the spatial scale and role of nonlinear effects

To quantify the sensitivity of the proposed methodology on the resolution of terrain data, we aggregate the MC-simulated radiation fields and corresponding terrain variables fields at increasing spatial scales ($L_s \in \{0.5, 1, 2, 3, 5, 10\}$ km), and in each case the predictive models are fit as described in Sect. 2.5, comparing the relative performance of multiple linear regressor and random forest predictions.

As shown in Fig. 7, the difference between RFR and MLR prediction skills is most relevant for the direct and direct-reflected flux components in the case of low solar zenith angles, and are also significant in the case of diffuse and coupled fluxes. In these two latter cases, the difference between RFR and MLR skills persists for all $\mu_0$ values and is thus not limited to cases where the Sun is relatively low on the horizon, as is the case for direct and direct-reflected fluxes. The reflected diffuse CE2 component is quite linear, as shown by the small difference between random forest and linear regressor predictive skills. For the direct-reflected flux this only happens for large enough values of the cosine of the solar zenith angle ($\mu_0 > 0.55$).

The predictive model for the direct flux shows high values of coefficients of determination ($R^2$), with similar performance for RFR and MLR, indicating that a linear model is well suited to describe this quantity. The only discrepancy is observed for very low solar angles ($\mu_0 = 0.1$), a case in which RFR outperforms the MLR. We believe this is primarily due to the effect of completely shaded areas in the domain, which are characterized by sharp transitions better described by an ensemble of decision trees due to the nonlinear behavior. We note that this effect is relevant only for comparatively small spatial scales ($L_s < 3$ km) and low solar angles and similarly impacts the direct-reflected flux, which as expected is modulated by the amount of direct light received at the surface. In the case of larger spatial averaging scales and larger solar angles, the MLR describes the direct flux with great accuracy. Moreover, the poorer model performance at the low solar angles is mitigated by the fact that these conditions (e.g., dusk and dawn) generally account for a small fraction of the irradiance received by land over most geographic locations and times of the year.

Reflected direct and reflected diffuse fluxes also exhibit a clear linear dependence on the terrain predictors, with MLR and RFR having similar $R^2$ values at all averaging scales and solar angles, with the only difference between the two approaches again appearing for the reflected direct flux for very small solar angles.

Appreciable differences between MLR and RFR are observed for diffuse and coupled fluxes. In these cases, it appears that diffuse radiation is better described by a nonlinear model, as is the case for irradiance originating from multiple reflections at the ground and atmospheric scattering ($f_{coup}$). In these cases, consistent with previous findings by (Lee et al., 2011), the predictive ability of linear models is lower.

These out-of-sample results were obtained using training data from the Peru domain and testing data from the eastern Alps domain. For completeness, we also ran the opposite configuration (switching training and testing domains) and found similar results.

In all cases in which predictive skills of MLR and RFR diverge, we observe that when comparing in-sample and out-of-sample performance the loss of predictive skill is larger in the case of RFR. This finding is not surprising. Given the additional model complexity of the RFR approach with respect to MLR, our analysis confirms that it is more prone to overfitting the calibration dataset. Once this overfitting tendency is accounted for, our analysis selects the MLR as model of choice since applications of the methodology do inevitably require extrapolation of the results to new domains.

Based on these results, we generally recommend the adoption of the linear regression models at least for direct and reflected fluxes, given the good performance and model simplicity. Applications of RFR are in principle possible in ESMs, and it has been shown here to have good predictive performance for this specific problem. However, this comes at the cost of a lower interpretability, and based on the present analysis here RFR is not the model of choice, given the limited increase in predictive skill with respect to MLR, especially when tested in cross validation.

## 3.3 Nonlinearity and effect of averaging for the direct flux

The direct flux component is characterized by a nonlinear behavior in the case of completely shaded areas.

This behavior should be taken into account in our model, as averaging terrain properties over tiles with varying characteristic size and spatial configuration would lead in general to changes in the average predicted $f_{dir}$ for a given tile if model predictions are averaged over areas that include partial shades in the high-resolution true field. One possible way to capture this behavior is to predict $f_{dir}$ first over an entire grid cell to obtain its average value and then impose the rule CE4 that tile-by-tile predictions must match the average value of the direct flux correction over the entire grid cell ($\langle \tilde{f}_{dir} \rangle$). This can be achieved through the following transformation for a generic tile $i = 1, \ldots n_t$. We define the corrected value $\tilde{f}_{dir}^{(i)}$ as

$$\tilde{f}_{dir}^{(i)} = \langle f_{dir} \rangle + \left( \langle \tilde{f}_{dir} \rangle - \tilde{f}_{dir}^{(min)} \right) \frac{f_{dir}^{(i)} - \langle f_{dir} \rangle}{\langle f_{dir} \rangle - f_{dir}^{(min)}}. \tag{12}$$

This correction was obtained by imposing the rule that the $\tilde{f}_{dir}$ values predicted by the model for each tile conserve the grid-cell-average value, by correcting the original value $\langle f_{dir} \rangle = \sum_{i=1}^{n_t} p_i f_{dir}^{(i)}$, with $p_i$ the fractional area of

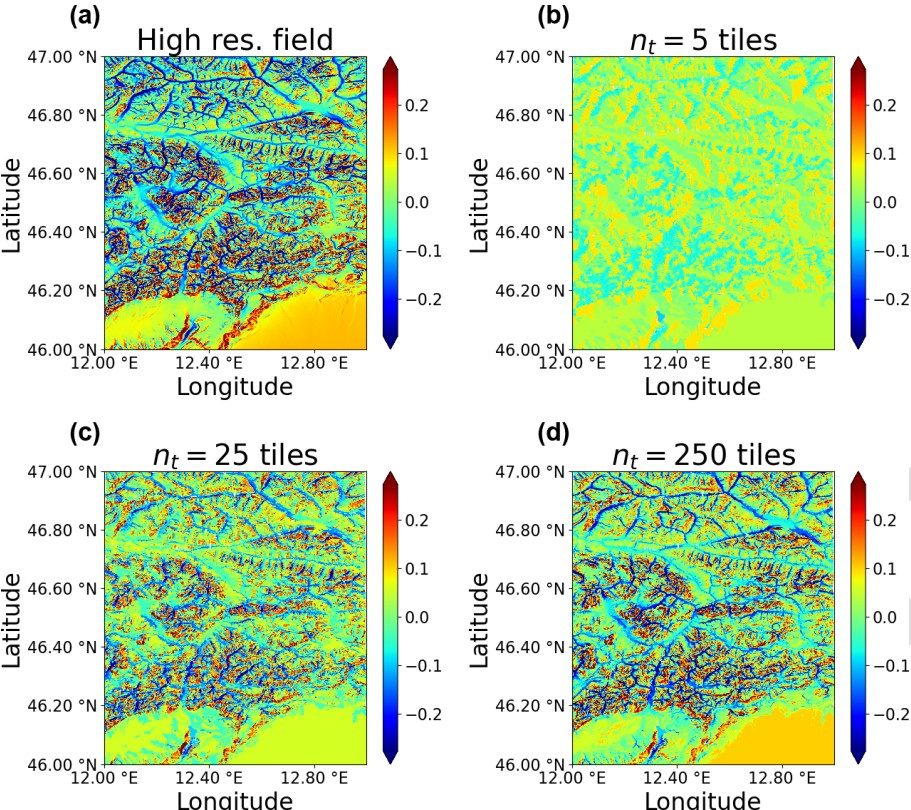

**Figure 11.** Normalized differences between 3D and PP diffuse fluxes for high-resolution predictions and predictions for different numbers of hillslopes ($k = 5, 25, 250$) setting a fixed number of tiles and hillslopes ($k = 5$). CE3

the grid cell assigned to tile $i$. This transformation also preserves the minimum value over the grid cells, meaning that $\tilde{f}_{\mathrm{dir}}^{(\mathrm{min})} = f_{\mathrm{dir}}^{(\mathrm{min})} = \min_{i=1}^{n_t} f_{\mathrm{dir}}^{(i)}$. For other flux variables this correction is not necessary if a linear model is used for predicting their average values over tiles (as done here).

Figure 8 shows the frequency distribution of corrections for the various flux terms that vary in their number of tiles. In particular, for the direct flux the distribution of tile-by-tile estimates is shown to converge with the histograms of the full high-resolution results for $n_{\mathrm{t}} = 250$ tiles. Thus, application of the correction given in Eq. (12) allows us to preserve the grid-cell-average correction while adequately representing the sub-grid-scale variability of $f_{\mathrm{dir}}$.

### 3.4 Model sensitivity to the number of tiles used

The predictive model can be used to produce tile-by-tile estimate of flux differences and compare results with those predicted for the original high-resolution terrain map. We repeat this analysis for different configurations of the hierarchical clustering scheme to test the sensitivity of the results to the number of tiles used to characterize domain heterogeneity. While the MC simulations were performed over two domains only, here we perform the clustering analysis over all three domains, comparing predictions obtained by applying the 3D

radiation corrections to the original high-resolution terrain data with the same approach to the sub-grid tiling structures for a varying number of land clusters. We expect a larger number of tiles to lead to a better representation of the flux component over rugged terrain. However, a number of tiles that is too high would not be feasible for running ESMs over large domains or the entire globe.

An example of the spatial distribution of tiles obtained by applying the HMC algorithm is reported in Fig. 9 for the eastern Alps domain, where the land domain is partitioned into 5 and 25 clusters. Two additional tiles are used to represent lakes and glaciers, which are present over this domain, albeit accounting for a small fraction of the surface. Once the tiled grid is defined using the HMC method, local averages of terrain parameters can be computed directly over each land unit. For example, the lower panels in Fig. 9 show the spatial distributions of the terrain view factor $\tilde{C}_t$) averaged over each tile, showing that the larger number of tiles greatly improves the representation of the spatial variability of topography.

We then tested the ability of the tiled grid when reproducing the actual spatial distribution of solar radiation by first examining the results obtained by separately varying the number of characteristic hillslopes $k$ (while keeping $p$ fixed) and comparing the results with the results for variable $p$ and

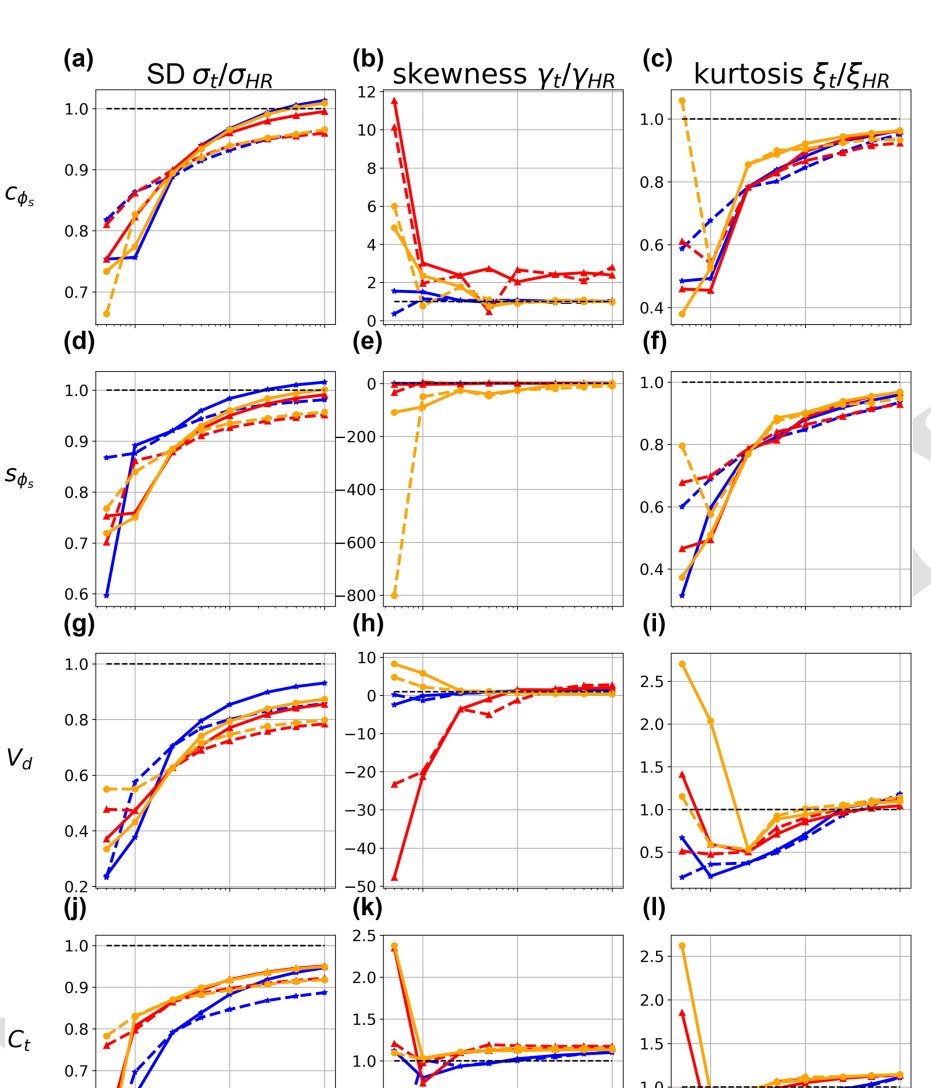

**Figure 12.** Convergence to the high-resolution terrain properties (slope orientation, sky view, and terrain view) obtained by increasing the number of tiles used in the parameterization. Two different tiling schemes are shown, obtained by fixing the number of hillslopes ($k = 5$) and varying that of sub-grid units $p$ (continuous lines) or by fixing $p = 5$ and varying $k$ (dashed lines). Results are reported for terrain variables.

fixed $k$. This analysis is intended to test the robustness of the method to different sub-grid land partitioning schemes.

Figures 10 and 11 show how increasing the number of land tiles improves the description of the direct flux and diffuse flux components, respectively, over the eastern Alps domain. Note that the same disaggregation of the domain in tiles is used for predicting the distribution of both variables. Even in the case of a fairly low number of tiles (e.g., five tiles in Figs.10b and 11b) the sub-grid structure is able to capture

the main feature of both direct and diffuse radiation fields, even though their variations are known to be controlled by different terrain properties (primarily aspect and sky view, respectively, as can be seen by the spatial distribution of direct and diffuse fluxes in the high-resolution results). For both $f_{\mathrm{dir}}$ and $f_{\mathrm{dif}}$, the tile-by-tile predictions appear to converge to the original high-resolution field when increasing the number of tiles (results for 25 and 250 tiles are shown in the lower panels of Figs. 10 and 11). Note that this result is quite sig-

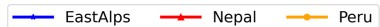

**Figure 13.** Convergence to the high-resolution radiation fields (direct, diffuse, reflected direct, and reflected diffuse differences between 3D and PP cases). Results are obtained by increasing the number of tiles used in the parameterization. Two different tiling schemes are shown obtained by fixing the number of hillslopes ($k = 5$) and varying that of sub-grid units $p$ (continuous lines) or fixing $p = 5$ and varying $k$ (dashed lines). Results are reported for normalized flux differences.

nificant, since for the domain examined here the number of points needed to obtain the high-resolution field without using a clustering approach would be on the order of $10^6$ (for a $1° \times 1°$ grid cell at the native 90 m resolution of SRTM data).

To obtain a more quantitative description of the convergence to the high-resolution fields, we show how spatial statistics of the radiation fields (spatial standard deviation $\sigma_x$, skewness $\gamma_x$, and kurtosis $\xi_x$) vary with increasing tile count

with respect to the same statistics computed for the reference high-resolution field.

For all the terrain predictors (Fig. 12) we find that when increasing the number of tiles, the spatial variability of terrain predictors converges to that of original high-resolution fields as expected. However, this convergence appears to be faster for the solar incidence angle when compared to $\tilde{V}_d$ and $\tilde{C}_t$. Higher-order statistics of the spatial fields (skewness and kurtosis) also tend to converge to the high-resolution field

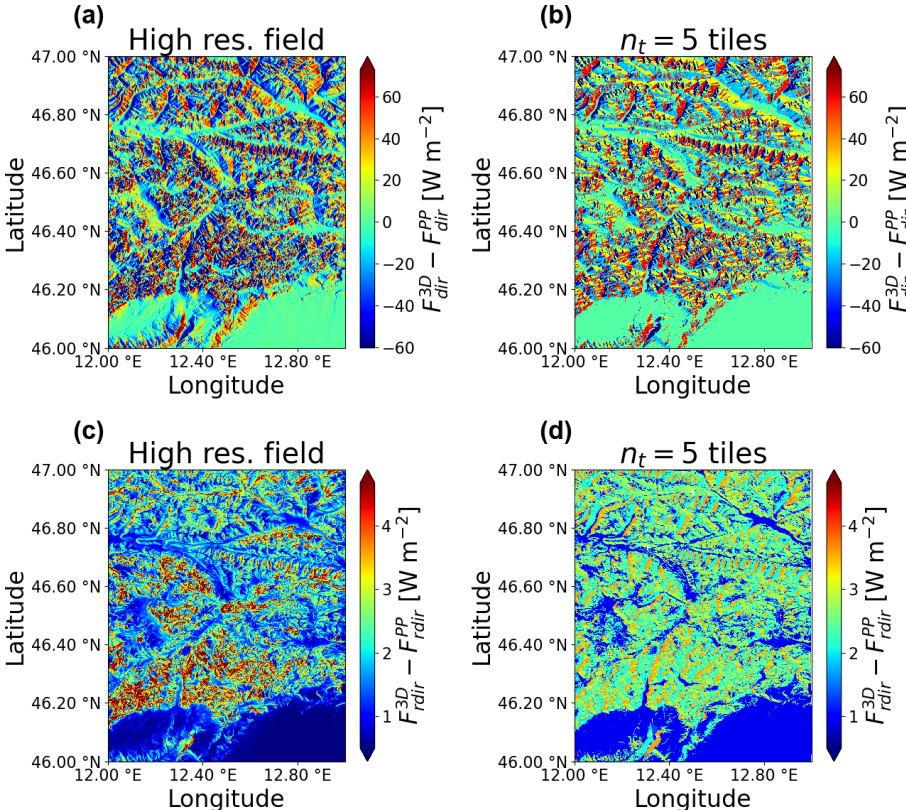

**Figure 14.** Downward radiation differences between 3D and PP fluxes for high-resolution predictions **(a, c)** and predictions using five tiles **(b, d)** for each flux component. Results are shown for a representative case of clear-sky conditions, and $\cos\theta_0 = 0.4$ is used for direct flux **(a, b)** and reflected direct flux **(c, d)**.

values, albeit with a larger variability. While spatial standard deviation is generally used as a metric for assessing spatial variability, examining skewness and kurtosis helps to make sure the entire distribution of tile-by-tile results converges to the high-resolution benchmark, since they better capture asymmetry and extremes in tile values. However, we note that these metrics are not very meaningful for the smallest number of tiles shown in Fig. 12) due to the small sample size. However, the fact that for a large enough number of tiles (e.g., $n_t > 20$) these metrics appear to converge to the true values increases our confidence that the hierarchical clustering scheme provides a good description of the topography heterogeneity.

For the flux variables (Fig. 13) we find a similar behavior, with convergence of the spatial standard deviation being generally faster than that of higher-order statistics. In this case, the convergence is faster for the direct flux and slower for all other flux component, as expected since $F_{\text{dir}}$ is primarily controlled by $\tilde{\mu}_i$, while the other flux components show a relevant dependence on either the sky view factor or terrain configurations.

To further analyze the configuration of the tiling structure used, we also tested different tiling configurations obtained by fixing the number of characteristic hillslopes ($k = 5$) and varying the number of lower-level land units in each hillslope ($p$) or conversely varying $k$ with $p = 5$ fixed. Results from both of these approaches are reported in Figs. 12 and 13. We find that generally convergence is faster using a larger $p$, i.e., dividing each characteristic slope in a larger number of tiles as opposed to increasing the number of characteristic hillslopes. This is not surprising. However, differences are generally small, and therefore the model proposed appears flexible and can in principle be applied with tiling that has been predefined in order to also accommodate for other physical processes.

### 3.5 Magnitude of predicted fluxes over varying atmospheric conditions

The presence of clouds adds considerable complexity to the problem of radiation–topography interactions and has not been considered in our work. A complete understanding of 3D land–atmosphere interactions would require us to extend our analysis to a large range of atmospheric conditions, which would be computationally costly, requiring a large number of ray-tracing simulations, and would arguably lead to a more complex parameterization requiring a larger number of parameters to estimate 3D topographic radiation cor-

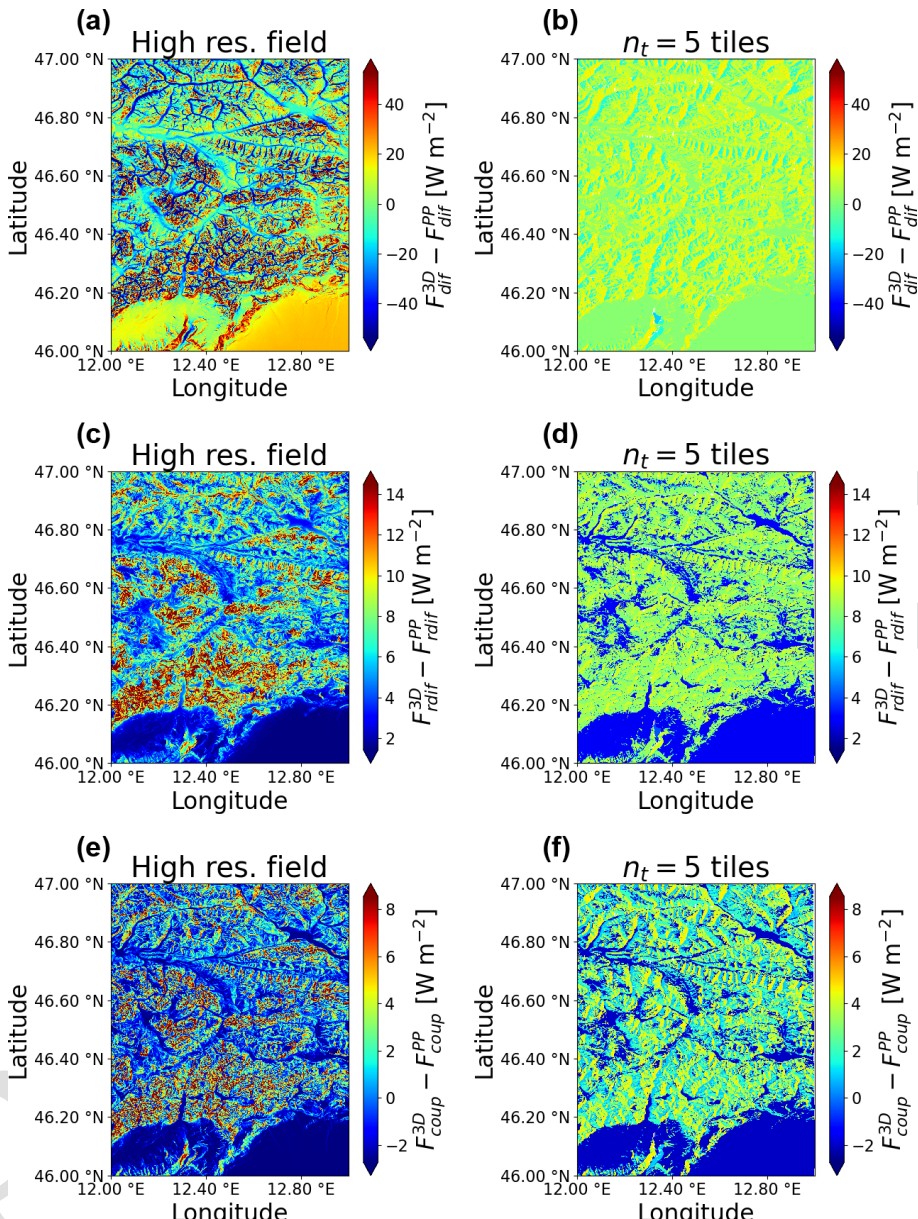

**Figure 15.** The same as Fig. 14 for diffuse, reflected diffuse, and coupled flux components.

rections. Previous work such as (Lee et al., 2011) also focused on clear-sky conditions. Although obtained assuming fixed atmospheric properties, the 3D terrain corrections for radiation fluxes have been formulated in dimensionless form (Eq. 1) so that they can be applied as a first-order correction to radiation received by land over varying atmospheric conditions. In different atmospheric conditions (clear vs. cloudy sky), the relative magnitude of the five radiation flux components can vary substantially. To show the magnitude of these changes, here we compute the magnitude of the flux 3D effects in dimensional form for the case of "clear sky" (i.e., aerosols but not clouds) and "total sky", i.e., atmospheric column with cloud cover and aerosol. These computations

were made using the Fu–Liou radiative transfer scheme (Fu and Liou, 1992) using a standard midlatitude summer atmospheric profile.

Results for clear-sky conditions are shown in Figs. 14 and 15, while the case of cloudy sky is shown in Fig. 16. While the overall downward flux is smaller in the case of cloudy sky, the direct and reflected direct fluxes are zero, meaning that the entire downward flux is comprised of diffuse, reflected diffuse, and coupled components. For the coupled flux, the spatial variations are similar in the two cases, with the most frequent values in the range $-2$ to $8\,\mathrm{W\,m^{-2}}$. For diffuse fluxes the clear-sky case is characterized by larger mag-

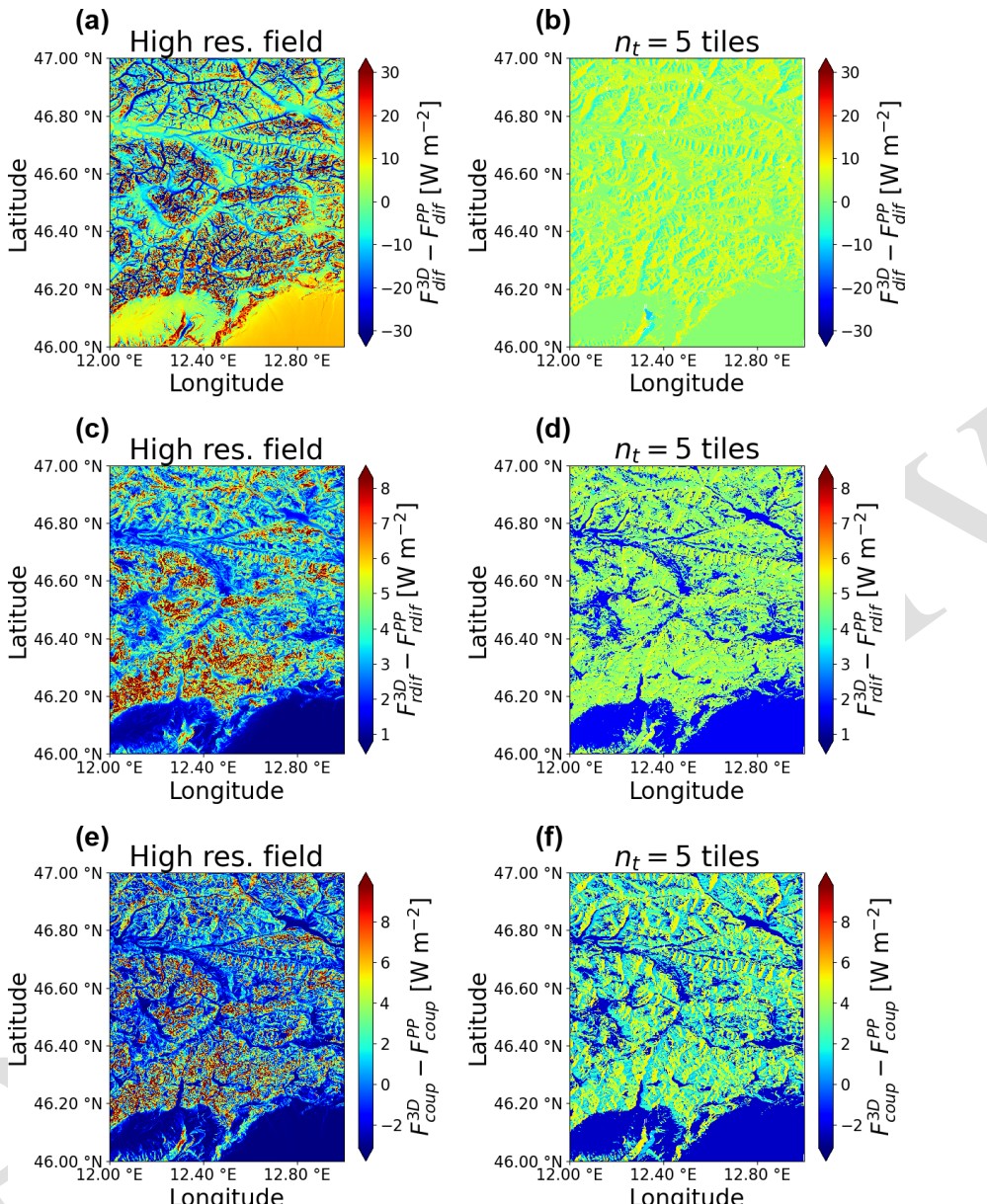

**Figure 16.** Downward radiation differences between 3D and PP fluxes for high-resolution predictions **(a, c, e)** and predictions using five tiles **(b, d, f)** for each flux component. Results are shown for a representative case of cloudy-sky conditions and $\cos\theta_0 = 0.4$. Direct and reflected direct fluxes are zero in this case.

nitude of 3D topographic effects, and this difference is even more marked in the case of reflected diffuse fluxes.

### 3.6 Compatibility with existing sub-grid tiling schemes

The approach proposed in this paper to design a sub-grid structure was developed keeping in mind the necessity of describing not only radiation but also other physical processes at the sub-grid scale. The GFDL model, as with several current-generation ESMs, resolves each sub-grid tile as a single "column" coupled with the atmosphere. Therefore, these

tiles should be flexible enough to meet the constraints posed by different physical processes. In our case, a single-level terrain clustering would suffice for the purpose of parameterizing 3D radiation–topography interactions. However, the multi-level clustering used here is flexible enough to accommodate multiple physical processes. For example, the outer-level clustering (i.e., the partition of the domain in $k$ characteristic hillslopes) is designed to obtain hydrologically coherent units so that processes like runoff and groundwater flow can be resolved in each homogeneous land unit (Chaney et al., 2018). Here we include this flexible sub-grid struc-

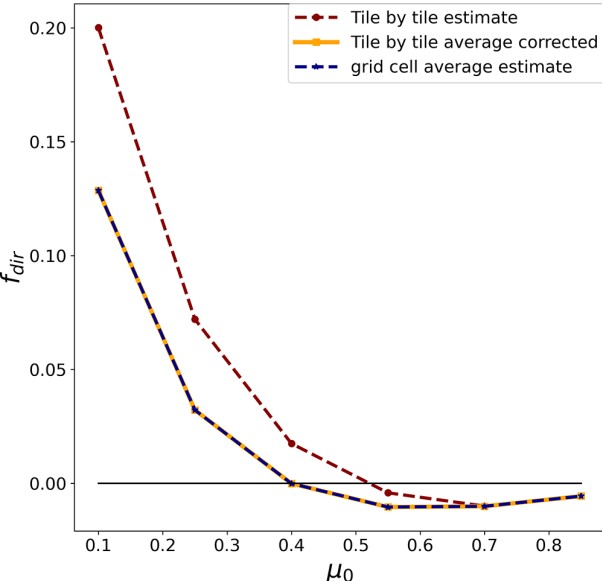

**Figure 17.** Comparison between grid average estimates of $f_{\mathrm{dir}}$ obtained by averaging the tile-by-tile corrections and by applying the predictive model for $f_{\mathrm{dir}}$ to the grid-cell-average terrain parameter predictors. Results are shown for the eastern Alps site for $\mu_0$ varying from 0.1 to 0.85 and solar azimuth $\phi = \pi/2$.

ture, and in our sensitivity study (Fig. 13) we compare different specifications of $k$ and $p$ (number of characteristic hillslopes and number of inner clusters within each hillslope, respectively) to test the sensitivity of our parameterization
to these changes in sub-grid structure, finding that our results are robust to the specific tile structure selected. Finally, it is possible that other variables may need to be added to the clustering to account for sub-grid heterogeneity of other processes (e.g., land use or soil properties). This can be di-
rectly done with the framework used here at the price of an increase in the number of tiles used. Producing an effective global-scale model grid able to meet these demands is possible but requires inevitable tradeoffs. The analysis in this paper contributes to this effort by quantifying the number of
tiles needed over mountainous terrain for the sole purpose of capturing the spatial variability in shortwave radiation. We note that in the case of the GFDL ESM, the model infrastructure is already suited for grids of this type, which can be characterized by an uneven number of sub-grid units in dif-
ferent grid cells depending on the local terrain properties. At the model start, the land grid cells can be distributed among available processors based on the estimated workload needed for each of the cells, assuming that computational cost is proportional to the number of sub-grid units. Therefore, the
work per processor is roughly the same, and the imbalance resulting from this uneven grid structure is minimized.

### 3.7 Comparison with grid-cell-average correction

The parameterization developed here can in principle be applied both to high-resolution terrain partitioned into tiles and to the original grid cell area. However, our analysis of
30 the model goodness of fit (Fig. 7) shows that the predictive models at different spatial resolutions (i.e., spatial averaging scales) show varying performance that improves with spatial scale. Therefore, according to this analysis, the model predictions for 3D effects over an entire grid cell have the best
performance. However, this comes at the cost of losing information about the sub-grid scales. Based on these considerations, it would be desirable to have a model with the best accuracy over an entire grid cell that is at the same time able to describe tile-by-tile spatial variations in radiation fluxes
received at the surface. In order for this to be possible, the parameterization developed here should be able to yield the same results when (i) applied to each tile separately and then averaged over the grid cell and (ii) applied to the grid average terrain predictors. We found that the parameterizations
proposed here have this property for four flux components (diffuse, coupled, and reflected fluxes) but not for the direct flux. This is because the behavior of the parameterization in the case of completely shaded areas (for which $f_{\mathrm{dir}}$ assumes the lower bound value $-1$) is not constant across averaging
scales due to the varying area of hill shades. We show this behavior in Fig. 17, where once averaged over an entire grid cell, the results of the tile-by-tile parameterization for the direct flux appear to overestimate the grid-cell-average 3D topographic effect with respect to the grid-cell-average es-
timate of $f_{\mathrm{dir}}$. This discrepancy can be resolved by the use of Eq. (12) which, once applied to the direct flux, allows us to impose the tile-by-tile correction to match the grid-cell-average correction. The result of this correction is shown in Fig. 17. Therefore, once this correction is employed for
the direct flux, this approach can at the same time conserve the area-average effect while providing information about the sub-grid variability of the radiative fluxes.

### 3.8 Discussion of the dependence on albedo and its spatial variability

The methodology described here has been developed for a surface with a uniform and fixed albedo value. For applying the methodology to an ESM, the radiation corrections which are albedo dependent (reflected and coupled fluxes) should be evaluated for the specific surface albedo value as
discussed in Sect. 2.5. However, in the presence of surfaces with spatially varying reflectivity, it is inevitable that different land tiles in the same grid cell will be characterized by different albedo values. For example, this can happen in the case of partial snow cover or in the case of different veg-
etation types along a gradient in elevation. While here the 3D radiation correction is applied to each tile independently, it is expected that reflected and coupled fluxes should in-

clude contributions from nearby tiles characterized by different albedo values. At present our methodology does not explicitly address these interactions between tiles with different albedo values. We note here two main difficulties in constructing such a model. First, its validation would require an extensive set of costly MC simulations to train the model over surfaces with varying reflectivity. Second, explicitly accounting for the reflectivity of nearby slopes may require a more complex statistical model, and it would not be easy task given the complex geometry of the tile structure adopted in our work. Solving these two important challenges should be object of future research.

## 4 Conclusions

Here we describe a methodology to compute solar fluxes over mountainous terrain, accounting for the sub-grid variability of topographic properties within a characteristic ESM grid cell. Topographic parameters modulating the incident solar irradiance, combined with results from Monte Carlo radiation simulation over 3D surface, are used to train a predictive model and cluster land surface based on topography–radiation interactions. The methodology as presented here is tailored to a tiling structure scheme recently introduced in the GFDL land model LM4.1 to describe the heterogeneity of hydrological properties. However, we believe this approach could be suitable for applications to other land surface models, as the clustering technique used here allows for a parsimonious description of the spatially varying solar fluxes. For this reason, we tested the sensitivity of the approach to the number of tiles used over independent sites characterized by complex topography. The results appear consistent over different geographical domains and indicate that even a limited number of tiles can reproduce a significant fraction of the spatial variability observed in the high-resolution fields. This result is particularly relevant when compared with standard approaches focused on increasing the land model resolution without adopting a clustering-based approach to construct a sub-grid land structure. Increasing the number of tiles improves not only the representation of spatial variances but also the convergence of higher-order statistics. However, even when using a lower number of tiles, the results remain consistent with previously developed grid-cell-average corrections (i.e., the methodology can ensure that the grid-cell-average correction is downward radiation is conserved) and thus are to be considered an improvement with respect to current plane-parallel radiative transfer. Here we found that even a limited number of tiles ($n_t = 10$) recovers a large fraction ($> 60\%$) of the spatial variance of irradiance over high-elevation mountain domains. However, we find that a larger number of sub-grid units (on the order of $n_t = 100$) would lead to a further significant improvement. Further increasing the number of tiles above $n_t = 100$ would lead to more modest improvements at the price of a much larger number of tiles required, as shown by the convergence analysis. Therefore, an optimal number of tiles could be between 10 and 100. Using such a large number of tiles in a global model would be ambitious at present due to its computational cost. However, we note that a global grid can be constructed by coarsening the sub-grid tile structure in an area with little or no topography and using more tiles in areas of complex terrain. Following this approach, constructing a global grid with a global average number of tiles between 5 and 20 (over land $1° \times 1°$ grid cells) is certainly in reach.

The current methodology, as well as previous studies on the topic (e.g., Chen et al., 2006; Lee et al., 2011), employed Monte Carlo simulations based on clear-sky conditions. Studying the effects of aerosols and cloud on the radiative transfer over complex terrain remains an open research avenue. In particular, the presence of spatially varying cloud cover could profoundly influence the spatial distribution of irradiance over mountainous terrain. However, in addition to the numerical challenges connected to the radiative transfer problem, including a spatially varying cloud cover would inevitably increase the number of parameters needed to parameterize the radiation received by the surface, thus posing a relevant parameterization challenge.

Studying the relationship between terrain predictors and irradiance differences between 3D and PP cases allowed us to quantify the importance of nonlinear effects and the relative skill of linear models and random forest predictors in capturing these relationships. We found that nonlinear effects are relevant primarily at the finer spatial scales and decrease drastically with spatial averages at increasing spatial scales. This result is consistent with previous investigations at coarser spatial scales based on linear models.

Based on our simulation study over a set mountainous domains, we quantified the difference in downward fluxes originating from topographic effects with respect to those obtained from traditional plane-parallel radiative transfer schemes for varying numbers of sub-grid tiles. Our results support the implementation of this methodology in the GFDL ESMs, which will be pursued as the next step to evaluate the effects of this correction of water and energy fluxes at the surface.

*Code and data availability.* The code used in this project is available in a Zenodo repository with the following DOI: https://doi.org/10.5281/zenodo.7714735 (Zorzetto, 2022a). The preprocessing software used to construct the land database for the GFDL land model v4.1 is included as supplementary material in the Zenodo repository https://doi.org/10.5281/zenodo.7720281(Zorzetto, 2023). A model dataset necessary to run the analysis is available in a Zenodo repository with the following DOI: https://doi.org/10.5281/zenodo.6975857 (Zorzetto, 2022b).

*Author contributions.* All authors contributed to research design. EZ developed the software, performed the simulations, and drafted the first version of the manuscript. All authors contributed to the manuscript.

*Competing interests.* The contact author has declared that none of the authors has any competing interests.

*Acknowledgements.* Lucas Harris and Linjiong Zhou are acknowledged for their comments and suggestions during an internal review of the manuscript CE5.

*Financial support.* This research has been supported by the National Aeronautics and Space Administration, Earth Sciences Division (grant no. 80HQTR21T0015) and the National Oceanic and Atmospheric Administration (grant no. NA18OAR4320123).

*Review statement.* This paper was edited by Mohamed Salim and reviewed by two anonymous referees.

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

**Remarks from the language copy-editor**