# Peer review of "Effects of complex terrain on the shortwave radiative balance: A sub-grid scale parameterization for the GFDL Land Model version 4.1"

_EGUsphere, 2022_

## Author Comment (AC2)

**Manuscript MS No.: egusphere-2022-770**

**Response to reviewers**

We thank the two anonymous referees for the attention devoted to our work. We report in the following a point-by-point response to the comments from the reviewers. Our own comments (in black unformatted text) are reported following referee comment *(in blue)*.

On behalf of all authors,

Enrico Zorzetto

**Referee #1**

*Topography controls many land surface processes. This manuscript combined an existing parameterization for solar radiation over complex terrain with a novel hierarchical multivariate clustering algorithm in GFDL. This work is very interesting and promising for applying in land surface models. However, how the authors considered the land cover types with different albedo values and energy balance is not clear; the performance of the proposed tile-level methods against the original grid-cell level methods for calculating regional average values is unknown; and more details in the physical explanations of some equations needs to be clarified,. Besides, how will the authors combine their tile separating and the existing tile schemes in GFDL? Please see below for my specific comments.*

We thank this reviewer for their comments. See the point-by-point response below.

*Major comments:*

1. *Line108-109: the authors stated that in GFDL, the diffuse radiation received by the (flat) surface corresponds here to the sum of Fdif and Fcoup. If so, how did the authors calculate Fdif and Fcoup in Eq.1 for GFDL?*

   This is not a new issue for 3D radiation studies as most atmospheric models provide as a boundary condition to the land model Fdif as the sum of these two terms. The recommended way to obtain this quantity is to compute from the atmospheric model an additional estimate of Fdif by imposing a completely black surface (so that Fcoup=0 as there is no surface reflection) and then pass both values to the land model, so that Fcoup can be obtained from their difference. We will explicitly discuss this approach in the revised manuscript version.

In this work, the proposed parametrization for these two radiation flux terms was compared with Monte Carlo simulations in which they could be calculated exactly by tagging each photon after an atmospheric scattering event or after one or more than one reflection at the ground, so that the results shown in the paper are consistent in how Fdif is defined, so that Fdif shown and defined in the paper does never include Fcoup.

2. *Eq. 1 and line 260: how did the authors consider the land surface with different albedo (e.g., snow and vegetation)? Different land cover types may have different albedo and thus different reflected energy from adjacent terrain.*

In the analysis performed in this paper the surface albedo is assumed to be uniform over each domain. While this is not a realistic assumption, we believe at this stage this was a necessary one. Assuming uniform albedo allowed us to isolate the effects of topography, summarized in the set of predictors used in the predictive model developed here. Using non uniform albedo would considerably increase the complexity of the problem. First, a spatially variable albedo would have required a much more extensive set of Monte Carlo experiments, necessary to sample a wide enough range of surface conditions (i.e., combinations of local topographic features and surface reflectivity values). Second, in developing the statistical model linking terrain predictors to 3D radiation effects, the presence of spatially varying albedo would have greatly increased the dimensionality of the problem and the number of predictors needed. For example, in addition to the terrain view factor at each point, some additional measure accounting for the reflectivity of adjacent slopes visible from a target point would be needed. While this can be done for a point, defining such measure for our land tiles would have been quite complex, since they have variable configuration and geometry. Moreover, such a measure would need to be time-varying based on the condition of the surface (e.g., presence of vegetation, snow cover etc., would need to be considered at each model time step to compute the topographic correction) On the other hand, including surface albedo as a clustering variable to create land tile is not particularly difficult. We also note that in the current model formulation, the topographic corrections for reflected fluxes are indeed albedo-dependent, so that when applied to the surface are scaled by the albedo (However, as noted above, in the present study the albedo value is spatially uniform).

We will add these considerations to the revised manuscript. We believe this is an important point worth of future investigation, but difficult to pursue for the reasons stated here.

3. *Eq. 2: here the irradiance Ek,l is based on horizontal plane or the inclined plane of pixel k,l? Please give more physical explanations for equation 2.*

In the revised manuscript we will further clarify the meaning of eq. (2), reported here:

$$E_{k,l} = E_0 \cos\theta_0 \frac{1}{N} \frac{A}{A_{k,l}} \sum_{i=1}^{N_{k,l}^{(s)}} w_i$$

This equation is used to compute the energy $E_{k,l}$ received by each cell of the land surface model, identified by the indices k,l. While a 3D mesh is used in the Monte Carlo simulations, the area of the cell $A_{k,l}$ is defined as the area of the cell on the horizontal plane, for consistency with the definition of area in the land model where the method will be applied, while A is the horizontal area of the entire domain and N the total number of incident photons. The equation computes the radiation received by a single land surface cell as a fraction of the radiation flux at the top of the surface $E_0$ by summing the "energy packets" $w_i$ of the photons absorbed over that area. Since the interactions of each photon are tagged (e.g., atmospheric scattering and / or previous reflections at the surface) the radiation received can be classified in one of the 5 flux components as defined in the paper. We will clarify the description of the equation to include these details.

4. *Line 264-266: please give more details about the energy conservation and albedo modification.*

We agree with the comment and will add some additional comments on energy conservation and albedo modifications. We did not add much detail on this in our original submission because the approach is described in Lee et al., 2015, but we agree that this is relevant for applications. The reason for this correction is that, by correcting fluxes received by land due to topography, energy is not necessarily conserved within a single model column. This is for example the case due to non-local effects: some model grid cells may in general receive overall more or less radiation due to their average topographic properties. This is in general accompanied by changes in the radiation received by neighboring grid cells. Moreover, in the case of reflections, the surface receives in general more energy with respect to the case of a flat surface. Properly accounting for these local and non-local effects is challenging in current ESMs, in which each land grid cell is directly coupled with the atmosphere but not directly with nearby model columns. A way to ensure that 3D radiation effects can be accounted for was proposed by Lee et al., 2015: In this approach, an effective "3D albedo" is computed for each land

model grid cell, such that a land grid—cell characterized by this "3D albedo" and forced by plane-parallel radiation (PP) absorbs the same amount of radiation as in the case of a surface characterized by actual land albedo (PP), forced by the 3D-corrected downward radiation fluxes. By returning the 3D albedo (which effectively represents the reflectivity of a "rough" land surface) to the atmosphere, energy is conserved while accounting for the 3D topographic correction.

*5. Line 273-274: will their difference be larger for cloudy condition?*

We note that the presence of clouds adds considerable complexity to the problem of radiation-topography interactions, and has not been considered in our work. Previous studies, in particular Lee et al., 2011, also based their work on clear-sky condition in order to make the problem more manageable. The main limitation would be the number of Monte Carlo simulations to run to sample different atmospheric conditions, and the increasing number of parameters in the statistical model used to estimate 3D topographic radiation corrections. However, given the importance of cloud cover we will add the following comments when discussing the limitations of our work.

It is quite possible that in cloudy conditions the differences between 3D and PP radiation fields will be different than in the case of clear sky analyzed here. However, we would expect the largest differences to arise in case of non-homogeneous cloud cover over the domain. This is a very difficult problem to model, as the number of configurations of 3D clouds and topography would be difficult to manage.

Even In the case of homogeneous cloud cover, we would expect a change of the relative magnitude of the 5 flux components, although the effect on 3D corrections proposed here remains unknown. We welcome future work focusing on this important issue.

*6. Figure 5: why can sky view factors be larger to 1?*

The reason is that what we are plotting here is the ratio of the sky view factor (a number between 0 and 1) to the cosine of the local terrain slope (smaller or equal to one), so that their ratio can be larger than unity. This quantity is defined in the paper using a tilde and is used as a predictor in the statistical model. However, we will add a comment on this point to make this definition clearer for the reader.

*7. Eq. 12: is this method only empirical?*

The equation is not empirical. The correction in eq. (12) was obtained by imposing the grid--cell average of the quantity of interest is conserved within the specified domain, while at the same time preserving its physical lower bound. We will clarify this in the manuscript.

8. How about the performance of the proposed tile-level methods against the original grid-cell level methods for calculating regional average values?

We agree that comparing the performance of the sub—grid parameterization with the regional average value is important. We will add a comparison between grid—average model and sub—grid estimates in the revised manuscript. However, we note that the analysis reported in Figure 7 already goes in this direction, as it tests the predictive model against "ground truth" Monte Carlo simulations for increasing values of spatial averaging scale. We note here that at small scales, performance increases with averaging scale. Therefore, it is easier for the model to capture the average effect over at a large enough scale. At large enough scale (say, above 10km) the approach should effectively become equivalent to a grid—cell average prediction, depending on the grid cell scale.

On the other hand, we note here that the main advantage of capturing the sub—grid distribution of irradiance can be appreciated only when examining the effects on a model run. This Is true especially for variables that are nonlinearly related to shortwave radiation, such as e.g., snow cover and land surface temperature. In this case, we believe that using a refined sub-grid distribution of solar radiation, while keeping its average value constant, can lead to non-zero grid average effects. This should be the object of a follow-up study, running a land model with both sub—grid and grid average models.

We will add these considerations to the revised version of the manuscript.

9. The authors presents the results based on the corrected factors in Eq.1. However, they may be not easy to understand. How about presenting some results about the radiation fluxes directly, which will be more clearer for the readers?

We agree with this comment and will add these results in the revised version of the manuscript. We have shown dimensionless quantities since the model produces corrections in this form. However, dimensional corrections can be directly obtained from these and we agree that including them in the presentation would improve the physical insight from our results.

10. The authors proposed tile-level topographic correction methods for solar radiation over complex terrain. However, current sub-grid tile schemes in GFDL consider different soil and vegetation, and topographic characteristics for simulating water and carbon cycles. How did the authors merge their clustering methods for radiation and the existing scheme in GFDL for other processes?

The approach proposed in our manuscript was developed keeping in mind the necessity of describing other physical processes at the sub—grid scale. In particular, the fact that the clustering is hierarchical is not strictly necessary for the purpose of solar radiation-topography interactions. A single--level terrain clustering would suffice for this purpose. However, the multi-level clustering used here accounts for the need for other land processes. For example, the outer level clustering (i.e., the partition of the domain in k characteristic hillslopes) is done to obtain hydrologically coherent units (for an example of their application to study soil moisture, we refer the reader to Chaney et al., 2018). We retain this flexible sub—grid structure, and in our sensitivity study (Figure 13 in the manuscript) we compare different specifications of k and p (number of characteristic hillslopes, and number of inner clusters within each hillslope) to test the sensitivity of our parameterization to these changes in sub-grid structure. Finally, it is likely that other variables may need to be added to the clustering to account for sub—grid heterogeneity of other processes. For example, soil properties. This can be directly done with the framework used here, at the price of an increase in the number of tiles used. Producing an effective global--scale model grid able to meet these demands is possible but requires some tradeoffs. The analysis in this paper contribute to this effort by quantifying the number of tiles needed over mountainous terrain for the sole purpose of capturing the spatial variability in shortwave radiation.

Minor comments:

1. Line 16-18: It will be better to show some quantitative metrics rather than only descriptive expression.

We agree with this comment. We will report in the abstract the main quantitative findings from our work: In particular, the magnitude of local topographic effects in our domain with respect to grid average estimates, and the number of sub—grid units necessary to represent sub—grid heterogeneity in downward fluxes.

2. Line 42: why did the author call this method 'WLH'?

In the revised version of the manuscript, this method will be called LLH following the initials of the authors of the Lee et al., 2011 paper.

3. Line 60-85: these summarize the objective and work of this paper. I suggest the authors simplify them for making them clearer.

We agree with the suggestion and will simplify this part of the introduction providing a better summary of our objectives.

4. Line124-125: Citing the corresponding papers may be better.

The papers cited here were not properly formatted. We will revise the formatting of this paragraph in the revised version, including the correct citations.

5. Line 93: how about vegetation with different PFTs?

Vegetation is certainly relevant for this problem, since it significantly modulates land albedo. Vegetation was not explicitly considered in our study, except to the extent to which it contributes to the albedo of each land tile. In our work, albedo is indeed accounted for to estimate the magnitude of reflected fluxes (see our response to major comment #2). However, vegetation PFTs and land use are not used to cluster the domain in our study. For the clustering, we focused on topographic quantities derived from the digital elevation model which are known to modulate irradiance over mountains. Surface albedo, vegetation and land use are certainly relevant variables for the problem at hand, and including them in the set of variables used to cluster land is in principle possible. However, the main benefit of doing so would stem from being able explicitly track radiation reflected between pairs of tiles: i.e., explicitly considering the albedo of nearby slopes visible from a tile, instead of computing reflected fluxes base on a uniform albedo value. While appealing, this approach would greatly increase the dimensionality of the problem and lead to a much more complex parameterization for reflected fluxes. One additional complication is the potentially time varying land use and vegetation structure which is not considered in our static definition of land—sub--grid structures.

6. Line 207: kp -> k*p?

Agree, will revise as suggested

7. *Line 413: he -> the*

Agree, will revise as suggested

**Referee #2**

**Summary and general comments**

*In this study, a parameterisation for the effects of sub-grid topography on surface shortwave radiation is presented. In a first step, the authors apply Monte Carlo ray tracing to simulate surface shortwave radiation for 3 geographic domains with complex terrain. These experiments serve as a reference to develop the (sub-grid) parametrisation. In a next step, terrain properties (μ, sky view factor and terrain configuration) are linked to modulated radiation fluxes with two statistical models – a Multiple Linear and a Random Forest Regression. Finally, sub-grid effects are considered by merging land units within a grid cell with similar terrain properties by means of hierarchical clustering.*

*The aim of this study is very interesting and relevant – namely improving the representation of surface shortwave radiation fluxes in an Earth System Model. Due to the plane parallel radiative transfer schemes applied in such models, surface radiation is typically simulated rather inaccurately in areas with complex terrain. The implementation of parameterisations,*

*particularly on a sub-grid scale, has the potential to strongly reduce such biases. The approach presented by the authors is very interesting and the manuscript is well written and structured. However, I struggled to understand certain sections in detail – for instance the hierarchical clustering section in the methods and some passages in the Results and Discussion. Furthermore, the Results and Discussion section is sometimes incomplete in my opinion and should be extended (see the following comments for more details).*

**Major comments**

**Section about hierarchical clustering (2.4)**

Until section 2.4, the methodology is very well described. However, I struggled to follow section 2.4. For instance, why do you want to partition land in hydrologically coherent units? From a "terrain-radiation-perspective" – this is not obvious. Has this approach been chosen due to an already existing tile classification in the GFDL Land Model?

This approach we followed has been selected to be compatible with the existing structure of the GFDL land model. Many physical processes other that radiation-topography interactions benefit from the sub-grid structure. However, using a different sub-grid partition for different processes does not appear to be a viable solution, due to considerable increase in model complexity and computational expense. For this reason, land processes are solved in a column for each land tile, using as boundary condition downward fluxes corrected based on 3D topography.

*We report here a response to reviewer #1 which addresses this point:*

> The approach proposed in our manuscript was developed keeping in mind the necessity of describing other physical processes at the sub—grid scale. In particular, the fact that the clustering is hierarchical is not strictly necessary for the purpose of solar radiation-topography interactions. A single--level terrain clustering would suffice for this purpose. However, the multi-level clustering used here accounts for the need for other land processes. For example, the outer level clustering (i.e., the partition of the domain in k characteristic hillslopes) is done to obtain hydrologically coherent units (for an example of their application to study soil moisture, we refer the reader to Chaney et al., 2018). We retain this flexible sub—grid structure, and in our sensitivity study (Figure 13 in the manuscript) we compare different specifications of k and p (number of characteristic hillslopes, and number of inner clusters within each

hillslope) to test the sensitivity of our parameterization to these changes in sub-grid structure. Finally, it is likely that other variables may need to be added to the clustering to account for sub—grid heterogeneity of other processes. For example, soil properties. This can be directly done with the framework used here, at the price of an increase in the number of tiles used. Producing an effective global--scale model grid able to meet these demands is possible but requires some tradeoffs. The analysis in this paper contribute to this effort by quantifying the number of tiles needed over mountainous terrain for the sole purpose of capturing the spatial variability in shortwave radiation.

I'm also confused why the clustering is performed twice (first in $k$ hillslopes, then in $p$ sub-units). I think a detailed flow diagram (e.g. with an example of the step-wise classification of sub-units of a geographic domain) would help the reader to understand these steps. Furthermore, it is also not obvious to me why lakes and glaciers represent separate classes. And are glaciers and lake classes further divided into sub-classes according to their terrain properties? Finally, some parts of section 3.4 (e.g. starting from line 354 could also be moved to the method section).

The reason for the hierarchical clustering is precisely that it must accommodate other physical processes other that radiation. For example, the subdivision in hillslopes is suitable to hydrological studies (e.g., Chaney et al., 2018). See our detailed response to the previous comment. We agree that a workflow diagram would help the reader understand our work and will include one in the revised manuscript.

In the current GFDL land model structure, lake and glacier are treated as separate land classes (solved as a separate "vertical columns"). In our work, land is subdivided in tiles using the hierarchical clustering scheme, while glacier and lake are treated each as a single tile, each characterized by average topographic properties.

However, we note that in our study domains glacier and lakes constitute a small fraction of the total area. There is no reason why glacier and lakes could not also be subdivided in multiple tiles if they occupy a relevant portion of the domain. This would be comparable to what we have done here, since the only difference for the purpose of our study would be the reflectivity of the surface. Our results here would equally apply to the case in which lake and glaciers were also partitioned in multiple clusters, since for the purposes of this study these three land classes behave the same way.

**Analysis and results – improve consistency and completeness**

- I'm missing the third domain (Nepal) in Fig. 7. I guess you used one domain to train the model and the other two domains for cross-validation – right?

Due to the considerable computational expense of ray tracing simulations, we performed these analyses for two domains only (for a large number of solar angles) and used them for training and testing. Three domains are then used to construct clustering and evaluate cluster-by-cluster results with high-resolution results.

- I think a performance comparison of the sub-grid to a grid-scale parameterisation would be very interesting to show. With this, you could emphasize the additional benefit of the sub-grid scale scheme.

We agree that this comparison would be useful, as also pointed out by referee #1. See our comment below:

We agree that comparing the performance of the sub—grid parameterization with the regional average value is important. We will add a comparison between grid—average model and sub—grid estimates in the revised manuscript. However, we note that the analysis reported in Figure 7 already goes in this direction, as it tests the predictive model against "ground truth" Monte Carlo simulations for increasing values of spatial averaging scale. We note here that at small scales, performance increases with averaging scale. Therefore, it is easier for the model to capture the average effect over at a large enough scale. At large enough scale (say, above 10km) the approach should effectively become equivalent to a grid—cell average prediction, depending on the grid cell scale.

On the other hand, we note here that the main advantage of capturing the sub—grid distribution of irradiance can be appreciated only when examining the effects on a model run. This Is true especially for variables that are nonlinearly related to shortwave radiation, such as e.g., snow cover and land surface temperature. In this case, we believe that using a refined sub-grid distribution of solar radiation, while keeping its average value constant, can lead to non-zero grid average effects. This should be the object of a follow-up study, running a land model with both sub—grid and grid average models.

We will add these considerations to the revised version of the manuscript.

- The discussion of certain findings should be extended. From the results, it seems that a tile number of ~100 captures the sub-grid characteristics already very well. Do you agree? And would such a number be feasible in an online ESM simulation?

We agree with the referee that this is a very important point, since one of the main purposes of the clustering technique used here is to make this problem tractable at the global scale.

The fact that about 100 clusters capture a significant fraction of the spatial distribution of irradiance is a very encouraging result in our opinion. To date, it is unfeasible to run a global model with 100 tiles / grid cell. An average number of 5 to 10 tiles is possible, however. We note that the domains selected for this study are characterized by very complex terrain, but not all land areas are. Therefore, we would argue that a grid setup could be constructed ranging from say 20 to 50 tiles in high mountains areas to a few over flat areas would be feasible. In the revised paper, we provide an estimate and guideline for constructing such a grid. For example, this can be done by increasing linearly the number of tiles from a small number (say, 3) to a large number (say, 20) based on the local elevation standard deviation. Moreover, we note that reproducing the entire spatial variance of solar irradiance (the values to which plots in Figure 13 converge) is certainly a bold objective for a global scale model, and a more limited number of tiles would still be appealing when compared to using simple grid—cell average values.

**Minor comments**

**Content-related (text)**

**Line 42:** what does the abbreviation "WLH" stand for?

The abbreviation was revised to LLH, to indicate the initials of the authors of the study referenced here.

**L139:** "uniform" albedo -> how realistic is this assumption?

See also related comment to referee #1:

In the analysis performed in this paper the surface albedo is assumed to be uniform over each domain. While this is not a realistic assumption, we believe at this stage this was a necessary one. Assuming uniform albedo allowed us to isolate the effects of topography, summarized in the set of predictors used in the predictive model developed here. Using non uniform albedo would considerably increase the complexity of the problem. First, a spatially variable albedo would have required a much more extensive set of Monte Carlo experiments, necessary to sample a wide enough range of surface conditions (i.e., combinations of local topographic features and surface reflectivity values). Second, in developing the statistical model linking terrain predictors to 3D radiation effects, the presence of spatially varying albedo would have greatly increased the dimensionality of the problem and the number of predictors needed. For example, in addition to the terrain view factor at each point, some additional measure accounting for the reflectivity of adjacent slopes visible from a target point would be needed. While this can be done for a point, defining such measure for our land tiles would have been quite complex, since they have variable configuration and geometry. Moreover, such a measure would need to be time-varying based on the condition of the surface (e.g., presence of vegetation, snow cover etc., would need to be considered at each model time step to compute the topographic correction) On the other hand, including surface albedo as a clustering variable to create land tile is not particularly difficult. We also note that in the current model formulation, the topographic corrections for reflected fluxes are indeed albedo-dependent, so that when applied to the surface are scaled by the albedo (However, as noted above, in the present study the albedo value is spatially uniform).

We will add these considerations to the revised manuscript. We believe this is an important point worth of future investigation, but difficult for the reasons stated here.

**L139:** I appreciate such clear definitions, it simplifies the comprehensibility of the subsequent text greatly!

Thank you for the comment!

**L157:** I'm not sure if I understand this sentence correctly. Do you mean that radiation fluxes significantly departure **locally** from areal-average fluxes?

Precisely. This was clarified to make it clear for the reader what we mean.

**L 162:** "represents the fraction of the sky dome visible from a target site" -> technically, this is incorrect. Compare e.g. with Helbig et al. (2009) (text next to Eq. 8) and Zakšek et al. (2011). The sky view factor definition of Dozier and Frew (1990) yields the fraction of hemispherical radiation received under the assumption of isotropic radiation. The same is valid for the subsequent explanation of the terrain configuration factor $C_t$.

We agree with the comment, the sentence was not correct. We will rephrase stating that the sky view factor used here is defined as the ratio of diffuse sky irradiance at a point to that on an unobstructed horizontal surface, in case of isotropic diffuse radiation.

**L171:** Could you explain why you use this terrain configuration definition and not simply $C_t = 1.0 - V_d$ (compare e.g. with Chu et al., 2021)?

Thank you for pointing out this interesting reference.

According to Dozier and Frew (1990) and e.g., Chen et al. (2006) the relation between sky view factor and terrain configuration factor can be approximated to (See eq. 9b in Dozier and Frew 1990, or eq. 6 in Chen et al, 2006):

$$C_t = \frac{1}{\pi} \int_0^{2\pi} \int_{H_\phi}^{\psi_\phi} \sin\theta [\cos\theta \cos\theta_s + \sin\theta \sin\theta_s \cos(\phi - \phi_s)] d\theta d\phi$$
$$\approx \frac{1 + \cos\theta_s}{2} - V_d$$

The relation $C_t = 1.0 - V_d$ can be further obtained by assuming locally flat surface $(\cos\theta_s = 1)$, but in general does not hold for a sloping surface.

**L174:** I would briefly introduce and explain the parameters $\mu_i$ and $\mu_0$ here.

Agree, will revised as suggested

**L194:** I'm a bit confused by these lines. It seems that you perform the clustering only for soil elements (also according to line 207; kpand kp + 2) and not for glaciers and lakes. What is the reason behind this? I guess glaciated areas and lakes can also have very variable topographic parameters (like e.g. sky view factor).

Yes, that is correct. The reason for this is that in the current GFDL land model configuration, glacier and lakes are treated separately from "soil" land, and in the current work we cluster soil in hillslopes and clusters, but consider lakes and glaciers each as a single cluster characterized by their areal-average topographic properties.

However, we note that in our study domains glacier and lakes constitute a small fraction of the total area. There is no reason why glacier and lakes could not also be subdivided in multiple tiles if they occupy a relevant portion of the domain. This would be comparable to what we have done here, since the only difference for the purpose of our study would be the reflectivity of the surface. Our results here would equally apply to the case in which lake and glaciers were also partitioned in multiple clusters, since for the purposes of this study these three land classes behave the same way.

**L204:** It's not obvious to me why you apply the clustering a second time. Generally, to increase the comprehensibility of this section, it might be worth to extend the workflow diagram displayed in Fig. 4. One could show the classification of a certain domain (resolved for every single step).

*[see also following comment, and our response to major comment #1]* We agree that this point should be clarified in the text and will add a workflow diagram as suggested.

The main reason for the multi-level clustering is obtaining a flexible sub—grid structure able to be applied to other processes other that the topographic radiation correction pursued here. We discuss this in detail in the response to comment #1.

**L207:** I'm still a bit puzzled – what is the motivation behind categorizing land surface based on hydrological properties? I don't see the connection to topography-radiation-processes.

[*See also our response to the previous comment, and to major comment #1*]

**L298:** "reflected components are quite linear" -> for $f_{rdir}$, the deviations between MLR and RFR are quite substantial...

This is correct. We argue that this is due to the interaction of solar incident effects (relevant for direct incident light) and terrain configuration, which is relevant for reflected fluxes. We will point this out in the revised text and amend this sentence.

**L 305:** "case in which..." -> I don't understand this part; there is probably something missing.

We agree the sentence is unclear and will revise it as follows: *"In the case of larger spatial averaging scales and larger solar angles, the MLR describes the direct flux with great accuracy".*

**L316:** First of all, I'm confused about which region (East Alps vs. Peru) is the (in-)dependent domain. The caption of Fig. 7 does not agree with the statement here.

Furthermore, I'm not convinced that results from RFR are not location dependent. Looking at Fig. 7, the RFR method consistently indicates a worse performance for the cross-validation domain than the MLR method. For me, this is an indication that obtained relations from the RFR simulation are very location-dependent and not easily transferable to other terrain geometries (i.e. the model is overfitted).

As stated in the caption of Figure 7, here *"The models were trained over the Peru site and tested over the same site (SS, continuous lines) and over the independent EastAlps site for cross validation (CV, dashed lines). ".* At line 316 we meant to say that we did run both configurations (switching the training and testing sites) and obtained comparable results. We will make this clearer in the text and will include the other case in the online supplementary material for completeness.

We agree with the second part of this comment. The statistical models used "learn" from the input data and this is indeed the reason why the cross validation was performed. We think it is important to quantify this effect. Of course, for applications we are interested in extrapolating to different regions. For extrapolation, we agree that the in-sample performance is not representative. For this reason, the out-of-sample goodness of fit (which, while not being as good as the in-sample one, is not that bad) can be used to assess method performance. We also agree that RFR does appear to overfit the training data to some extent (exhibiting a larger difference between in- and out-of-sample cases with respect to MLR), and this is the reason why in the discussion we argue about the advantages of the linear model (MLR), despite some nonlinear behavior being observed for some of the flux components. Will make this tradeoff between the two models more explicit in the discussion section.

**L388:** It would be interesting to see the results for these tests too. Maybe you could show them in the supplementary material.

Agree. These results can be automatically generated running the code included with manuscript and will be included in a supplementary material in our revised submission.

**Typos, phrasing and stylistic comments**

**L124:** references not correctly rendered

Agree, will be revised as suggested

**L153:** I was a bit confused by this line, it might be better to write something like: "The MC calculations were performed for three independent domains (Nepal, Peru, East Alps)..." (if that is what you mean)

We run the Monte Carlo simulations for two domains only, as explained above due to the computational expense and the need to run for several solar angles. Three domains are used in the clustering study to get additional data points to evaluate the effect of the number of tiles, but the third domain is not used to train and test the predictive model for the radiation correction terms. We will clarify this important point to make it clear.

**L157:** "determines" -> "determine"

Agree, will revise as suggested

**L162:** "represent" -> "represents"

Agree, will revise as suggested

**L166:** "in order to compute **the** sky view factor"

Agree, will revise as suggested

**L198:** "eq. 6" -> "Eq. 6"

Agree, will revise as suggested

**L215:** "if these **are** present in a given grid cell."

Agree, will revise as suggested

**L220:** "**the** is the indicator" -> "is the indicator"

Agree, will revise as suggested

**L263:** "eqns. (1)" -> "Eq. (1)"

Agree, will revise as suggested

**L273:** "angles compute based" -> "angles compute**d** based"

Agree, will revise as suggested

**L273:** "simulation (5)" -> "simulation (Fig. 5)"?

Agree, will revise as suggested

**L290:** I would rewrite this to e.g.: "…larger than approximately 5 km the effect disappears."

Agree, will revise as suggested

**L302:** "case in which" -> "a case in which"

 Agree, will revise as suggested

**Figures and Tables**

**Figure 2:** The colorbar labelling is erroneous – I guess it should be "Elevation [m a.s.l.]". The same is true for the upper-left panel in figure 3. Furthermore, the degree symbol is missing for the cardinal directions.

The colorbar label refers to elevation above mean sea level (m.s.l.). We do not believe it is erroneous as reported in our submission. Elevation [m a.s.l.] as suggested by the reviewer would also be correct. However, we will clarify that we mean elevation above mean sea level to avoid any confusion.

**Figure 4:** It seems from these panels (x/y-coordinates) that the MC model was run on a map projection. Could you specify the projection somewhere?

Agree – it is an equal area Mollweide projection. We will include this information in the revised manuscript.

**Figure 7:** $\mu_0$ not correctly rendered in caption

 Will be revised as suggested

**New references**

Chu, Q., Yan, G., Qi, J., Mu, X., Li, L., Tong, Y., et al. (2021). Quantitative analysis of terrain reflected solar radiation in snow-covered mountains: A case study in Southeastern Tibetan Plateau. Journal of Geophysical Research: Atmospheres, 126, e2020JD034294. https://doi.org/10.1029/2020JD034294

Helbig, N., Löwe, H., & Lehning, M. (2009). Radiosity Approach for the Shortwave Surface Radiation Balance in Complex Terrain, Journal of the Atmospheric Sciences, 66(9), 2900-2912. https://doi.org/10.1175/2009JAS2940.1

Zakšek, K.; Oštir, K.; Kokalj, Å½. Sky-View Factor as a Relief Visualization Technique. Remote Sens. 2011, 3, 398-415. https://doi.org/10.3390/rs3020398

**References**

Lee, Wei-Liang, K. N. Liou, and Alex Hall. "Parameterization of solar fluxes over mountain surfaces for application to climate models." Journal of Geophysical Research: Atmospheres116.D1 (2011).

Lee, W-L., et al. "A global model simulation for 3-D radiative transfer impact on surface hydrology over the Sierra Nevada and Rocky Mountains." Atmospheric Chemistry and Physics15.10 (2015): 5405-5413.

Dozier, Jeff, and James Frew. "Rapid calculation of terrain parameters for radiation modeling from digital elevation data." IEEE Transactions on geoscience and remote sensing 28.5 (1990): 963-969.

Chaney, N. W., Van Huijgevoort, M. H., Shevliakova, E., Malyshev, S., Milly, P. C., Gauthier, P. P., & Sulman, B. N. (2018). Harnessing big data to rethink land

heterogeneity in Earth system models. Hydrology and Earth System Sciences, 22(6), 3311-3330.

---

## Author Response (AR1)

**Manuscript MS No.: egusphere-2022-770**

**Response to reviewers and description on the revised manuscript**

We thank the two anonymous referees for the attention devoted to our work. We report in the following a point-by-point response to the comments from the reviewers. Our own comments (in black unformatted text) are reported following referee comment *(in blue)*. Where necessary, excerpts of the revised text are reported. In the following, line numbers refer to the revised submission text unless explicitly stated.

The title has been updated to the following, to reflect the published version of the GFDL model this parameterization was developed for:

*"Effects of complex terrain on the shortwave radiative balance: A sub--grid scale parameterization for the GFDL land model version 4.1"*

On behalf of all authors,

Enrico Zorzetto

**Referee #1**

*Topography controls many land surface processes. This manuscript combined an existing parameterization for solar radiation over complex terrain with a novel hierarchical multivariate clustering algorithm in GFDL. This work is very interesting and promising for applying in land surface models. However, how the authors considered the land cover types with different albedo values and energy balance is not clear; the performance of the proposed tile-level methods against the original grid-cell level methods for calculating regional average values is unknown; and more details in the physical explanations of some equations needs to be clarified,. Besides, how will the authors combine their tile separating and the existing tile schemes in GFDL? Please see below for my specific comments.*

We thank this reviewer for their comments. Please see the point-by-point response below for a response to each specific comment. We have now revised the paper addressing the issues raise in the review process and we believe this process has significantly improved the manuscript.

*Major comments:*

1. *Line108-109: the authors stated that in GFDL, the diffuse radiation received by the (flat) surface corresponds here to the sum of Fdif and Fcoup. If so, how did the authors calculate Fdif and Fcoup in Eq.1 for GFDL?*

This discrepancy is an issue that our work has common with several other studies since it is common for coupled land - atmosphere models to provide the diffuse flux as the sum of Fdif and Fcoup according to the formalism used in our work. By doing so, information is lost about the land-reflected component of this flux. The recommended way to obtain this quantity is to compute from the atmospheric model an additional estimate of the diffuse radiative flux passed to the land component by imposing a completely black surface, so that in this case Fcoup=0). We now explicitly discuss this approach in the revised manuscript version at line

From the revised manuscript at line 104:

"For a flat surface $F_{rdir}$, $F_{rdif} = 0$ while $F_{dir}$, $F_{dif}$, and $F_{coup} \neq 0$ in general. We note that in the GFDL land model, diffuse radiation received by a flat surface with albedo $\alpha$ ( $F_{dif,[LM|\alpha]}$) corresponds here to the sum of $F_{dif}$ and $F_{coup}$. We note that these quantities can be computed separately by computing first the diffuse flux corresponding to a black surface $F_{dif,[LM|\alpha=0]}$, case in which the coupled flux is zero, and by then computing the coupled flux for the actual land surface as $F_{coup} = F_{dif,[LM|\alpha]} - F_{dif,[LM|\alpha=0]}$. Conversely, the diffuse flux can be obtained as $F_{dif} = F_{dif,[LM|\alpha=0]}$. Based on this formalism, the normalized flux differences between the traditional plane-parallel (PP) case and the topography-aware case (3D) are the quantities object of our analysis which can be used to correct the shortwave radiative balance in ESMs."

2. *Eq. 1 and line 260: how did the authors consider the land surface with different albedo (e.g., snow and vegetation)? Different land cover types may have different albedo and thus different reflected energy from adjacent terrain.*

It is correct that the topographic corrections for reflected fluxes are indeed albedo dependent. We have performed the analysis in our manuscript for a fixed albedo value. However, the method can be generalized to surfaces with any albedo. We have now clarified after introducing Eq. (1) that the quantities as defined here are albedo dependent. At line 115:

*"While direct and diffuse components are independent of surface albedo, the reflected flux components are linearly dependent on albedo. Finally, the coupled flux is nonlinearly dependent on surface albedo."*

Further at line 287 we discuss the issue of albedo dependence, and how predictions for reflected fluxes can be directly evaluated by rescaling them for the actual surface reflectivity. However, the coupled flux, which is nonlinearly dependent on albedo, would require e.g., interpolation for different albedo values, as proposed by Lee 2011 referenced here at line 287:

*"We note that these equations were derived for a single surface reflectivity value. The direct and diffuse flux components are independent on albedo; Reflected fluxes are linearly dependent on albedo so that predicted value can be rescaled by the surface albedo. Finally, $F_{coup}$ is nonlinearly dependent on the surface albedo so that predictions for different albedo values can e.g. be obtained by interpolation"*

Furthermore, In the analysis performed in this paper the surface albedo is assumed to be spatially uniform over each domain. While we agree this is not a realistic assumption, we believe in this work it was a necessary one. Assuming uniform albedo allowed us to isolate the effects of topography, summarized in the set of predictors used in the predictive model developed here. Using spatially non uniform albedo would considerably increase the complexity of the problem. First, a spatially variable albedo would have required a much more extensive set of Monte Carlo experiments, necessary to sample a wide enough range of surface conditions (i.e., combinations of local topographic features and surface reflectivity values). Second, in developing the statistical model linking terrain predictors to 3D radiation effects, the presence of spatially varying albedo would have greatly increased the dimensionality of the problem and the number of predictors needed.

We have added these considerations to the revised manuscript. We believe this is an important point worth of future investigation, but difficult to pursue for the reasons stated here (line 491):

*"The methodology described here has been developed for a surface with a uniform and fixed albedo value. For applying the methodology to an ESM, the radiation corrections which are albedo - dependent (reflected and coupled fluxes) should be evaluated for the specific surface albedo value as discussed in Section 2.5. However, in the presence of surfaces with spatially varying reflectivity, it is inevitable that different land tiles in the same grid cell will be characterized by different albedo values. For example, this can happen in the case of partial snow cover, or in the case*

*of different vegetation types along a gradient in elevation. While here the 3D radiation correction is applied to each tile independently, it is expected that reflected and coupled fluxes should include contributions from nearby tiles characterized by different albedo values. At present our methodology does not address explicitly these interactions between tiles with different albedo. We note here two main difficulties in constructing such a model: First, its validation would require an extensive set of costly MC simulations to train the model over surfaces with varying reflectivity. Second, explicitly accounting for the reflectivity of nearby slopes may require a more complex statistical model, and it would not be easy task given the complex geometry of the tile structure adopted in our work. Solving these two important challenges should be object of future research."*

3. *Eq. 2: here the irradiance Ek,l is based on horizontal plane or the inclined plane of pixel k,l? Please give more physical explanations for equation 2.*

In the revised manuscript we will further clarify the meaning of eq. (2), reported here:

$$E_{k,l} = E_0 \cos\theta_0 \frac{1}{N} \frac{A}{A_{k,l}} \sum_{i=1}^{N_{k,l}^{(s)}} w_i$$

This equation is used to compute the energy $E_{k,l}$ received by each cell of the land surface model, identified by the indices k,l. While a 3D mesh is used in the Monte Carlo simulations, the area of the cell $A_{k,l}$ is defined as the area of the cell on the horizontal plane, for consistency with the definition of area in the land model where the method will be applied, while A is the horizontal area of the entire domain and N the total number of incident photons. The equation computes the radiation received by a single land surface cell as a fraction of the radiation flux at the top of the surface $E_0$ by summing the "energy packets" $w_i$ of the photons absorbed over that area. Since the interactions of each photon are tagged (e.g., atmospheric scattering and / or previous reflections at the surface) the radiation received can be classified in one of the 5 flux components as defined in the paper.

We have clarified the equation as follows (and furthermore refer to Meyer's work where the MC approach Is described in detail) at line 151:

*If $E_0$ is the radiation incident at the TOA with a cosine of the zenith angle $\mu_0 = \cos\theta_0$, then the horizontal distribution of solar irradiance received by the land surface is given by (Meyer, 2009):*

$$E_{k,l} = E_0 \cos\theta_0 \frac{1}{N} \frac{A}{A_{k,l}} \sum_{i=1}^{N_{k,l}^{(s)}} w_i$$

*where $w_i$ is the energy of the i-th incident photon, N is the total number of photons tracked in the simulation over a domain with area A, and $N_{k,l}^{(s)}$ is the number of photons absorbed by the surface within grid cell k,l with area horizontal plane surface area $A_{k,l}$. Photon are released with unit energy at the TOA and lose a fraction of this energy through absorption in each atmospheric layer (Meyer et al., 2009) or through absorption at the ground. While a 3D mesh is used in the Monte Carlo simulations, the area of the cell $A_{k,l}$ is defined as the area of the cell on the horizontal plane, for consistency with the definition of area in the land model where the method will be applied. Equation (2) expresses the radiation received by a single land surface cell as a fraction of the radiation flux at the top of the surface $E_0$ by summing the energy $w_i$ of all the photons absorbed by the surface over that area. Since the interactions of each photon are tagged (e.g., keeping track of atmospheric scattering and any previous reflections at the surface) the radiation received can be classified in one of the 5 flux components as defined in Eq. (1).*

4. *Line 264-266: please give more details about the energy conservation and albedo modification.*

We agree with the comment and have added some additional comments on the issue of energy conservation and albedo modifications. However, since this albedo modification was not applied in the current work (which is focused on correcting the downward radiation flux), we provide a brief description of the methodology and refer to the relevant literature which could be relevant for future applications.

We did not add much detail on this in our original submission because the approach is described in Lee et al., 2015, but we agree that this may be relevant for future applications of the methodology. The reason for employing this correction is that, by correcting fluxes received by land due to topography, energy is not necessarily conserved within a single model column. This is for example the case due to non-local effects: some model grid cells may in general receive overall more or less radiation due to their average topographic properties. This is in general accompanied by changes in the radiation received by neighboring grid cells. Moreover, in the case of reflections, the surface receives in general more energy

with respect to the case of a flat surface. Properly accounting for these local and non-local effects is challenging in current ESMs, in which each land grid cell is directly coupled with the atmosphere but not directly with nearby model columns. A way to ensure that 3D radiation effects can be accounted for was proposed by Lee et al., 2015: In this approach, an effective "3D albedo" is computed for each land model grid cell, such that a land grid—cell characterized by this "3D albedo" and forced by plane-parallel radiation (PP) absorbs the same amount of radiation as in the case of a surface characterized by actual land albedo (PP),  forced by the 3D-corrected downward radiation fluxes. By returning the 3D albedo (which effectively represents the reflectivity of a "rough" land surface) to the atmosphere, energy is conserved while accounting for the 3D topographic correction.

We have added the following discussion at line 291:

*In general, simply applying a correction to the downward radiation received by land will not ensure energy conservation. This is expected in general, as some of the 3D topographic effects parameterized here would lead in general to energy fluxes between neighboring land model grid cells. A procedure was proposed by Lee et al., (2015) which can be used to address this issue. In this approach, an effective albedo $\alpha_{3D}$ is computed for each land grid cell, such that a grid—cell characterized by this $\alpha_{3D}$ and forced by plane-parallel radiation (PP) absorbs the same amount of radiation as in the case of a surface characterized by actual land albedo $\alpha$, while forced by the 3D-corrected downward radiation fluxes. By returning the 3D albedo (which effectively represents the reflectivity of a "rough" but flat land surface) to the atmosphere, energy is conserved while accounting for the 3D topographic correction.*

5. *Line 273-274: will their difference be larger for cloudy condition?*

We note that the presence of clouds adds considerable complexity to the problem of radiation-topography interactions and has not been considered in our work. To the best of our knowledge previous work also based their work on clear-sky assumption in order to make the problem more manageable. We believe the main challenge to explicitly include the effects of clouds is the number of Monte Carlo simulations needed to sample different atmospheric conditions, and the increasing number of parameters in the statistical model used to estimate 3D topographic radiation corrections. However, given the importance of cloud cover we now discuss the issue in the manuscript.

It is quite possible that in cloudy conditions the differences between 3D and PP radiation fields will be different than in the case of clear sky analyzed here. However, we would expect the largest differences to arise in case of nonhomogeneous cloud cover over the domain. This is a very difficult problem to model, as the number of configurations of 3D clouds and topography would be difficult to manage. However, we agree that this is an important point and should be explored in future research.

In addition to potential changes in 3D radiation effects related to cloud cover, different atmospheric conditions do also lead to a change to the absolute and relative relative magnitude of the 5 radiation flux components. To show the magnitude of these changes, we now provide an additional comparison of the magnitude of out 3D – PP radiation estimates applied in the case of a clear sky and cloudy sky atmospheric conditions.

We have added the following section in the manuscript discussing this issue (line 436):

*The presence of clouds adds considerable complexity to the problem of radiation-topography interactions and has not been considered in our work. A complete understanding of 3D land - atmosphere interactions would require to extend our analysis to a large range of atmospheric conditions, which would be computationally costly requiring a large number of ray tracing simulations, and would arguably lead to a more complex parameterization requiring a larger number of parameters to estimate 3D topographic radiation corrections. Previous work such as Lee et al., (2011) also focused on clear sky conditions. Although obtained assuming fixed atmospheric properties, The 3D-terrain corrections for radiation fluxes have been formulated in dimensionless form (Eq. (1)) so that they can be applied as a first order correction to radiation received by land over varying atmospheric condition. In different atmospheric conditions (clear vs. cloudy sky), the relative magnitude of the 5 radiation flux components can vary substantially. To show the magnitude of these changes, we compute here the magnitude of the fluxes in dimensional form for the case of clear sky (i.e., aerosols but not cloud) and "total sky", i.e., atmospheric column with cloud cover and aerosol. These computations were made using the Fu-Liou radiative transfer scheme (Fu et al., 1992) using a standard mid latitude summer atmospheric profile.*

*Results for clear sky conditions and are shown in Figure 14 and Figure 15, while the case of cloudy sky is shown in Figure 16. While the overall downward flux is smaller in the case of cloudy sky, the direct and reflected-direct fluxes are zero so that the entire downward flux is comprised of diffuse, reflected-diffuse and coupled components. For the coupled flux, the spatial variations are similar in the two cases, with the most frequent values in the range -2 to 8 W m$^{-2}$. For diffuse fluxes the clear sky case is characterized by larger magnitude of 3D topographic effects, and this difference is even*

*more marked in the case of reflected-diffuse fluxes.*

[Figure]

Figure 15: Same as figure 14 for diffuse, reflected-diffuse and coupled flux components.

[Figure]

*Figure 14: Downward radiation differences between 3D and PP fluxes for high resolution predictions (left column) and predictions and for 5 tiles (right column) for each flux component. Results are shown for a representative case of clear sky conditions and cos theta_0 = 0.4 for direct flux (top panels) and reflected-directed flux (bottom).*

6.  *Figure 5: why can sky view factors be larger to 1?*

The reason is that what we are plotting here is the ratio of the sky view factor (a number between 0 and 1) to the cosine of the local terrain slope (smaller or equal to one), so that their ratio can be larger than unity. This quantity is defined in the

paper using a tilde and is used as a predictor in the statistical model. We clarified this point in the manuscript to make this definition clearer for the reader (line 459):

*For the purpose of parameterizing solar fluxes, we divide these terrain variables by the local terrain slope obtaining the normalized variables $\tilde{\mu}_i = \mu_i / \cos\theta_s$, $\tilde{V}_d = V_d / \cos\theta_s$, and $\tilde{C}_t = C_t / \cos\theta_s$ as recommended by Lee et al., 2013. Note that while $V_d \in [0,1]$, the ratio $\tilde{V}_d = V_d / \cos\theta_s$ can sometimes be larger than 1.*

7. Eq. 12: is this method only empirical?

Strictly speaking the equation is empirical but was obtained by imposing that both the spatial average and the physical lower bound of the direct radiation field are conserved. We clarified this in the description of the equation, which now reads (line 574):

*This correction was obtained by imposing that the $\tilde{f}_{dir}$ values predicted by the model for each tile conserve the grid cell average value, by correcting the original value $\langle f_{dir} \rangle = \sum_{i=1}^{n_t} p_i f_{dir}^{(i)}$, with $p_i$ the fractional area of the grid cell assigned to tile i. This transformation also preserves the minimum value over the grid cells, so that $\tilde{f}_{dir}^{(min)} = f_{dir}^{(min)} = \min_{i=1,n_t} f_{dir}^{(i)}$. For other flux variable this correction is not necessary if a linear model is used for predicting their average values over tiles, as done here.*

8. How about the performance of the proposed tile-level methods against the original grid-cell level methods for calculating regional average values?

We agree that comparing the performance of the sub—grid parameterization with the regional average value is important. We now discuss the difference between tile-by-tile predictions and grid—average model in the revised manuscript. However, we note that also the analysis reported in Figure 7 is relevant to answer this question, as it tests the predictive model against "ground truth" Monte Carlo simulations for increasing values of spatial averaging scale. We note here that at small scales, performance increases with averaging scale. Therefore, it is easier for the model to capture the average effect over at a large enough scale.

We have the following considerations in the manuscript at line 472:

*The parameterization developed here can be in principle be applied both to high resolution terrain partitioned in tiles and to the original grid cell area. However, our*

*analysis of the model goodness of fit (Figure 7) shows that the predictive models at different spatial resolutions (i.e., spatial averaging scale) show varying performance, which increases with spatial scale. Therefore, according to this analysis, the model predictions for 3D effects over an entire grid cell have the best performance. However, this comes at the cost of losing information about the sub—grid scales. Based on these considerations, it would be desirable to have a model with the best accuracy over an entire grid cell, that is at the same time able to describe tile-by-tile spatial variations in radiation fluxes received at the surface. In order for this to be possible, the parameterization developed here should be able to yield the same results when i) applied to each tile separately and then averaged over the grid cell, and ii) applied to the grid average terrain predictors. We found that the parameterizations proposed here have this property for 4 flux components (diffuse, coupled and reflected fluxes) but not for the direct flux. This is because the behavior of the parameterization in the case of complete shade areas (for which $f_{dir}$ assumes the lower bound value -1) is not constant across averaging scales due to the varying area of hills hades. We show this behavior in Figure 17, where once averaged over an entire grid cell, the results of the tile-by-tile parameterization for the direct flux appears to overestimate the grid-cell average 3D topographic effect. This discrepancy can be resolved by the use of eq. (12) which, once applied to the direct flux, allows to impose the tile-by-tile correction to match the grid cell average correction. Result of this correction is shown in Figure 17. Therefore, once this correction is employed for the direct flux, this approach can at the same time conserve the area average effect while providing information about the sub--grid variability of the radiative fluxes.*

[Figure]

Figure 17: Comparison between grid average estimates of f_dir obtained by averaging the tile-by-tile corrections, and by applying the predictive model for f_dir to the grid-cell average terrain parameter predictors. Results are shown for the EastAlps site.

9. The authors presents the results based on the corrected factors in Eq.1. However, they may be not easy to understand. How about presenting some results about the radiation fluxes directly, which will be more clearer for the readers?

We agree with this comment. We believe it is useful to show dimensionless correction since the method will be potentially applied to a range of forcing and atmospheric conditions leading to different magnitude of the resulting fluxes. However, it is also important to show the magnitude of the corrections. For this reason, we have now added a section in which we present dimensional results for varying atmospheric conditions (e.g., a case of clear sky and case of cloudy sky) to show the range of the results under these conditions.

Please see this results in our answer to comment #5 above, and in the revised manuscript at line 435. An example of the newly added plots which present 3D corrections in dimensional form:

[Figure]

Figure 16: Downward radiation differences between 3D and PP fluxes for high resolution predictions (left column) and predictions and for 5 tiles (right column) for each flux component. Results are shown for a representative case of cloudy sky conditions and and $\cos \theta_0 = 0.4$. Direct and reflected direct fluxes are zero in this case.

10. The authors proposed tile-level topographic correction methods for solar radiation over complex terrain. However, current sub-grid tile schemes in GFDL consider different soil and vegetation, and topographic characteristics for simulating water and carbon cycles. How did the authors merge their clustering methods for radiation and the existing scheme in GFDL for other processes?

We have added the following section in the manuscript discussing this important issue (line 455):

*The approach proposed in our manuscript to design a sub--grid structure was developed keeping in mind the necessity of describing not only radiation, but also other physical processes at the sub—grid scale. The GFDL model, as several current generation ESMs, resolves each sub-grid tile as a single 'column' coupled with the atmosphere. Therefore, these tiles should be flexible enough to meet the constraints posed by different physical processes. In our case, a single--level terrain clustering would suffice for the purpose of parameterizing 3D radiation-topography interactions. However, the multi-level clustering used here accounts is flexible enough to accommodate multiple physical processes. For example, the outer level clustering (i.e., the partition of the domain in k characteristic hillslopes) is designed to obtain hydrologically coherent units, so that processes like runoff and groundwater flow can be resolved in each homogeneous land unit (Chaney et al., 2018). Here we include this flexible sub—grid structure, and in our sensitivity study (Figure 13) we compare different specifications of k and p (number of characteristic hillslopes, and number of inner clusters within each hillslope) to test the sensitivity of our parameterization to these changes in sub-grid structure, finding that our results are robust to the specific tile structure selected. Finally, it is possible that other variables may need to be added to the clustering to account for sub—grid heterogeneity of other processes. For example, land use or soil properties. This can be directly done with the framework used here, at the price of an increase in the number of tiles used. Producing an effective global--scale model grid able to meet these demands is possible but requires inevitable trade-offs. The analysis in this paper contribute to this effort by quantifying the number of tiles needed over mountainous terrain for the sole purpose of capturing the spatial variability in shortwave radiation.*

Minor comments:

1. Line 16-18: It will be better to show some quantitative metrics rather than only descriptive expression.

We agree with this comment. We have now included in the abstract the main quantitative finding (line 19):

*"..quantify the importance of the topographic correction for a varying number of terrain clusters and for different radiation terms (direct, diffuse, and reflected radiative fluxes) in order to inform the application of this methodology in different ESMs with varying sub-grid tile structure. We find that even a limited number of sub-- grid units such as 10, can lead to recovering more than 60% of the spatial variability of solar irradiance over a mountainous area."*

2. Line 42: why did the author call this method 'WLH'?

In the revised version of the manuscript this method has been renamed LLH, after the initials of the authors of the Lee et al., 2011 paper.

3. Line 60-85: these summarize the objective and work of this paper. I suggest the authors simplify them for making them clearer.

We agree with the suggestion and have simplified this part of the introduction, removing some of the more technical material (description of the Monte Carlo technique used) and moving it to the methods section.

4. Line124-125: Citing the corresponding papers may be better.

The papers cited here were not properly formatted in the original submission. We have now amended the formatting of this paragraph in the revised version, including the correct citations.

5. Line 93: how about vegetation with different PFTs?

Vegetation is certainly relevant for this problem, since it can significantly modulate land albedo. In turn, in the GFDL land model vegetation is dynamic and introducing our approach to 3D radiation is expected to impact the evolution of vegetation over tiles with different insolation.

Vegetation was not explicitly considered in our approach, except to the extent to which it contributes to the albedo of each land tile. In our work, albedo is indeed accounted for to estimate the magnitude of reflected fluxes (see our response to major comment #2). However, vegetation PFTs and land use are not used to cluster the domain in our study. For the clustering, we focused on topographic quantities derived from the digital elevation model which are known to modulate irradiance over mountains. Surface albedo, vegetation and land use

are certainly relevant variables for the problem at hand and including them in the set of variables used to cluster land is in principle possible. However, the main benefit of doing so would stem from being able explicitly track radiation reflected between pairs of tiles: i.e., explicitly considering the albedo of nearby slopes visible from a tile, instead of computing reflected fluxes base on a uniform albedo value. While appealing, this approach would greatly increase the dimensionality of the problem and lead to a much more complex parameterization for reflected fluxes.

6. Line 207: kp -> k*p?

   Agree, revised as suggested.

7. *Line 413: he -> the*

   Agree, revised as suggested.

**Referee #2**

**Summary and general comments**

*In this study, a parameterisation for the effects of sub-grid topography on surface shortwave radiation is presented. In a first step, the authors apply Monte Carlo ray tracing to simulate surface shortwave radiation for 3 geographic domains with complex terrain. These experiments serve as a reference to develop the (sub-grid) parametrisation. In a next step, terrain properties (μ, sky view factor and terrain configuration) are linked to modulated radiation fluxes with two statistical models – a Multiple Linear and a Random Forest Regression. Finally, sub-grid effects are considered by merging land units within a grid cell with similar terrain properties by means of hierarchical clustering.*

*The aim of this study is very interesting and relevant – namely improving the representation of surface shortwave radiation fluxes in an Earth System Model. Due to the plane parallel radiative transfer schemes applied in such models, surface radiation is typically simulated rather inaccurately in areas with complex terrain. The implementation of parameterisations, particularly on a sub-grid scale, has the potential to strongly reduce such biases. The approach presented by the authors is very interesting and the manuscript is well written and structured. However, I struggled to understand certain sections in detail – for instance the hierarchical clustering section in the methods and some passages in the Results and Discussion. Furthermore, the Results and Discussion section is sometimes incomplete in my opinion and should be extended (see the following comments for more details).*

We thank this reviewer for their comments. Please see the point-by-point response below for a response to each specific comment. We have now revised the paper addressing the issues raised in the review process and we believe this process has significantly improved the manuscript. We have revised the description of the hierarchical clustering approach, explaining more in depth why the choice of this model structure was made. The Results and discussion section was also extended, prompted by the comments from both referees.

**Major comments**

**Section about hierarchical clustering (2.4)**

Until section 2.4, the methodology is very well described. However, I struggled to follow section 2.4. For instance, why do you want to partition land in hydrologically coherent units? From a "terrain-radiation-perspective" – this is not obvious. Has this approach been chosen due to an already existing tile classification in the GFDL Land Model?

This approach we followed has indeed been selected to be compatible with the existing structure of the GFDL land model. The land model must describe the coupled land-atmosphere system, which includes many processes other than the radiative transfer examined in this work. Many physical processes other that radiation-topography interactions benefit from the sub-grid structure. However, using a different sub-grid partition for different processes does not appear to be a viable solution, due to considerable increase in model complexity and computational expense. For this reason, land processes are solved in a single column for each land tile, using in our case as boundary condition the downward fluxes corrected based on the local 3D topography.

We have now added a discussion of the reasons for our choice (line 455):

> The approach proposed in our manuscript to design a sub--grid structure was developed keeping in mind the necessity of describing not only radiation, but also other physical processes at the sub—grid scale. The GFDL model, as several current generation ESMs, resolves each sub-grid tile as a single 'column' coupled with the atmosphere. Therefore, these tiles should be flexible enough to meet the constraints posed by different physical processes. In our case, a single--level terrain clustering would suffice for the purpose of parameterizing 3D radiation-topography interactions. However, the multi-level clustering used here accounts is flexible enough to accomodate multiple physical processes.  For example, the outer level clustering (i.e., the partition of the domain in k characteristic hillslopes) is designed to obtain hydrologically coherent units, so that processes like runoff and groundwater flow can be resolved in each homogeneous land unit (Chaney et al., 2018). Here we include this flexible sub—grid structure, and in our sensitivity study (Figure 13) we compare different specifications of k and p (number of characteristic hillslopes, and number of inner clusters within each hillslope) to test the sensitivity of our parameterization to these changes in sub-grid structure, finding that our results are robust to the specific tile structure selected. Finally, it is possible that other variables may need to be added to the clustering to account for sub—grid heterogeneity of other processes. For example, land use or soil properties. This can be directly done with the framework

*used here, at the price of an increase in the number of tiles used. Producing an effective global--scale model grid able to meet these demands is possible but requires inevitable trade-offs. The analysis in this paper contribute to this effort by quantifying the number of tiles needed over mountainous terrain for the sole purpose of capturing the spatial variability in shortwave radiation.*

I'm also confused why the clustering is performed twice (first in $k$ hillslopes, then in $p$ sub-units). I think a detailed flow diagram (e.g. with an example of the step-wise classification of sub-units of a geographic domain) would help the reader to understand these steps. Furthermore, it is also not obvious to me why lakes and glaciers represent separate classes. And are glaciers and lake classes further divided into sub-classes according to their terrain properties? Finally, some parts of section 3.4 (e.g. starting from line 354 could also be moved to the method section).

The reason for the hierarchical clustering is precisely that it must accommodate other physical processes other that radiation. For example, the subdivision in hillslopes is suitable to hydrological studies (e.g., Chaney et al., 2018). We have added a discussion of this choice – reported in our detailed response to the previous comment.

We agree that a workflow diagram would help the reader understand our work and we have included one in the revised manuscript.

In the current GFDL land model structure, lake and glacier are treated as separate land classes (solved as a separate "vertical columns"). In our work, land is subdivided in tiles using the hierarchical clustering scheme, while glacier and lake are treated each as a single tile, each characterized by average topographic properties.

However, we note that in our study domains glacier and lakes constitute a small fraction of the total area. There is no reason why glacier and lakes could not also be subdivided in multiple tiles if they occupy a relevant portion of the domain. This would be comparable to what we have done here, since the only difference for the purpose of our study would be the reflectivity of the surface. Our results here would equally apply to the case in which lake and glaciers were also partitioned in multiple clusters, since for the purposes of this study these three land classes behave the same way. We now discuss this in the manuscript at line 211:

*The land fraction of the study sites, which are chosen to represent in size a typical ESM grid cell, is first divided in a maximum of three components: soil, glacier, lake. The soil fraction is then subdivided into a set of tiles characterized by homogeneous terrain properties relevant for capturing the effects of topography on radiative transfer. Additionally, lake and glacier*

*areas, where present, are treated as individual separate tiles. We note that in the domains selected for this study lake and glacier areas constitute a small fraction of the total grid cell area. When applying the methodology to areas where glaciers cover a large fraction of the grid cell, it may be useful to also partition glacier areas in multiple clusters. This can be done following the same methodology described here, since for the purpose of radiation-terrain interactions the only relevant parameters would be the average albedo of each cluster (land or glacier).*

Furthermore, we added the following comment on the experimental setup (line 234)

*"A natural test for the ability of the tiled grid of reproducing the actual spatial distribution of solar radiation can be performed as follows: We test here the results for multiple HMC configurations obtained by varying the number of characteristic hillslopes (k) as well as the number of land units within each characteristic hillslope (p). For illustration purposes, we consider two cases: a fixed value of k=5 and a varying p, and the opposite (varying k, setting p=5). This experiment leads to a set of grid configurations with a number of tiles per grid cell varying from 5 to 1000, with different weights given to the first level (partitioning of land in hillslopes) and the second level, in which each characteristic hillslope is further subdivided in p homogeneous land units contributing to the overall number of tiles $n_t$. This experiment thus elucidates the relative performance of the two different levels of the hierarchical clustering approach in capturing the spatial heterogeneity of the domain."*

Finally, some parts of section 3.4 (e.g. starting from line 354 could also be moved to the method section).

As suggested, we have moved this material to the methods section, and here we have added a shorter comment (line 397):

*We then tested the ability of the tiled grid of reproducing the actual spatial distribution of solar radiation by examining first the results obtained by varying separately the number of characteristic hillslopes k (while keeping p fixed) and comparing then with the results for variable p and constant k. This analysis is intended to test the robustness of the method to different sub--grid land partitioning schemes.*

**Analysis and results – improve consistency and completeness**

- I'm missing the third domain (Nepal) in Fig. 7. I guess you used one domain to train the model and the other two domains for cross-validation – right?

The statement about 2 domains here was correct - we indeed run the Monte Carlo simulations for two domains only, as explained above due to the computational expense and the need to repeat our analysis for several solar angles. Three domains are then used in the clustering study to get additional data points to evaluate the effect of the number of tiles, but the third domain is not used to train and test the predictive model for the radiation correction terms. In this part of the analysis, the parameterization (trained and tested previously) is applied separately to both the high-resolution digital elevation maps, and to the sub—grid tiling schemes for varying number of clusters, comparing the results obtained in these different configurations.

We have added the following comment to the manuscript (line 167):

*"While only two domains are used for MC calculations due to the computational expense of this procedure, an additional domain located in high mountain Asia (Figure 2) is used to further test the results of the spatial clustering over areas with different topographic features."*

And at line 385:

*"While the MC simulations were performed over two domains only, here we perform the clustering analysis over all three domains comparing predictions obtained by applying the 3D radiation corrections to the original high-resolution data with the same approach to the sub--grid tiling structures for a varying number of land clusters."*

- I think a performance comparison of the sub-grid to a grid-scale parameterisation would be very interesting to show. With this, you could emphasize the additional benefit of the sub-grid scale scheme.

We agree that comparing the performance of the sub—grid parameterization with the regional average value is important. We now discuss the difference between tile-by-tile predictions and grid—average model in the revised manuscript. However, we note that also the analysis reported in Figure 7 is relevant to answer this question, as it tests the predictive model against "ground truth" Monte Carlo simulations for increasing values of spatial averaging scale. We note here that at small scales, performance increases with averaging scale. Therefore, it is easier for the model to capture the average effect over at a large enough scale.

We have the following figure and discussion in the manuscript at line 472:

[Figure]

*"The parameterization developed here can be in principle be applied both to high resolution terrain partitioned in tiles and to the original grid cell area. However, our analysis of the model goodness of fit (Figure 7) shows that the predictive models at different spatial resolutions (i.e., spatial averaging scale) show varying performance, which increases with spatial scale. Therefore, according to this analysis, the model predictions for 3D effects over an entire grid cell have the best performance. However, this comes at the cost of losing information about the sub—grid scales. Based on these considerations, it would be desirable to have a model with the best accuracy over an entire grid cell, that is at the same time able to describe tile-by-tile spatial variations in radiation fluxes received at the surface. In order for this to be possible, the parameterization developed here should be able to yield the same results when i) applied to each tile separately and then averaged over the grid cell, and ii) applied to the grid average terrain predictors. We found that the parameterizations proposed here have this property for 4 flux components (diffuse, coupled and reflected fluxes) but not for the direct flux. This is because the behavior of the parameterization in the case of complete shade areas (for which $f_{dir}$ assumes the lower bound value -1) is not constant across averaging scales due to the varying area of hill shades. We show this behavior in Figure 17, where once averaged over an entire grid cell, the results of the tile-by-tile parameterization for the direct flux appears to overestimate the grid-cell average 3D topographic effect. This discrepancy can be resolved by the use of eq. (12) which, once applied to the direct flux, allows to impose the tile-by-tile correction to match the grid cell average correction. Result of this correction is shown in Figure 17. Therefore, once this correction is employed for the direct flux, this approach can at the same time conserve the area average effect while providing information about the sub--grid variability of the radiative fluxes."*

- The discussion of certain findings should be extended. From the results, it seems that a tile number of ~100 captures the sub-grid characteristics already very well. Do you agree? And would such a number be feasible in an online ESM simulation?

We agree with the referee that this is a very important point, since one of the main purposes of the clustering technique used here is to make this problem tractable at the global scale.

The fact that about 100 clusters capture a significant fraction of the spatial distribution of irradiance is a very encouraging result in our opinion. To date, it is unfeasible to run a global model with 100 tiles / grid cell. An average number of 5 to 20 tiles is possible, however. We note that the domains selected for this study are characterized by very complex terrain, but not all land areas are. Therefore, we would argue that a grid setup could be constructed ranging from say 20 to 50 tiles in high mountains areas to just a few over flat areas would be feasible. In the revised paper, we now provide an estimate and guidelines for constructing such a grid. For example, this can be done by increasing linearly the number of tiles from a small number (say, 3) to a large number (say, 20) based on the local elevation standard deviation. Moreover, we note that reproducing the entire spatial variance of solar irradiance (the values to which plots in Figure 13 converge) is certainly a bold objective for a global scale model, and a more limited number of tiles would still be appealing when compared to using simple grid—cell average values.

We have added the following discussion of this important issue (line 513):

*"This result is particularly relevant when compared with standard approaches focused on increasing the land model resolution without adopting a clustering-based approach to construct a sub--grid land structure. Increasing the number of tiles does not only improve the representation of spatial variances, but also improves convergence of higher order statistics. However, even when using a lower number of tiles, the results remain consistent with previously (i.e., the methodology can ensure that the grid cell average correction is downward radiation is conserved) developed grid-cell average correction and thus are to be considered an improvement with respect to current plane-parallel radiative transfer. Here we found that even a limited number of tiles $n_t = 10$) recovers a large fraction (> 60%) of the spatial variance of irradiance over high mountain domains. However, we find that a larger number of sub--grid units (of the order of $n_t = 100$) would lead*

*to a further significant improvement. Further increasing the number of tiles above*
*$n_t = 100$ would lead to more modest improvements at the price of a much larger*
*number of tiles required. Therefore, an optimal number of tiles could be between*
*10 and 100. Using such a large number of tiles in a global model would be*
*ambitious to date, due to its computational cost. However, we note that a global*
*grid could be constructed by coarsening the sub--grid number of tiles in area with*
*little or no topography. Following this approach, constructing a global grid with a*
*global average number of tiles between 5 and 20 is certainly in reach."*

**Minor comments**

**Content-related (text)**

**Line 42:** what does the abbreviation "WLH" stand for?

The abbreviation was revised to LLH, to indicate the initials of the authors of the study
referenced here.

**L139:** "uniform" albedo -> how realistic is this assumption?

> In the analysis performed in this paper the surface albedo is assumed to be
> uniform over each domain. While this is not a realistic assumption, we believe at
> this stage this was a necessary one. Assuming uniform albedo allowed us to
> isolate the effects of topography, summarized in the set of predictors used in
> the predictive model developed here. Using non uniform albedo would
> considerably increase the complexity of the problem. First, a spatially variable
> albedo would have required a much more extensive set of Monte Carlo
> experiments, necessary to sample a wide enough range of surface conditions
> (i.e., combinations of local topographic features and surface reflectivity values).
> Second, in developing the statistical model linking terrain predictors to 3D
> radiation effects, the presence of spatially varying albedo would have greatly
> increased the dimensionality of the problem and the number of predictors
> needed.

> We have added these considerations to the revised manuscript. We believe this
> is an important point worth of future investigation, but difficult to pursue for the
> reasons stated here (line 491):

*"The methodology described here has been developed for a surface with a uniform and fixed albedo value. For applying the methodology to an ESM, the radiation corrections which are albedo - dependent (reflected and coupled fluxes) should be evaluated for the specific surface albedo value as discussed in Section 2.5. However, in the presence of surfaces with spatially varying reflectivity, it is inevitable that different land tiles in the same grid cell will be characterized by different albedo values. For example, this can happen in the case of partial snow cover, or in the case of different vegetation types along a gradient in elevation. While here the 3D radiation correction is applied to each tile independently, it is expected that reflected and coupled fluxes should include contributions from nearby tiles characterized by different albedo values. At present our methodology does not address explicitly these interactions between tiles with different albedo. We note here two main difficulties in constructing such a model: First, its validation would require an extensive set of costly MC simulations to train the model over surfaces with varying reflectivity. Second, explicitly accounting for the reflectivity of nearby slopes may require a more complex statistical model, and it would not be easy task given the complex geometry of the tile structure adopted in our work. Solving these two important challenges should be object of future research."*

**L139:** I appreciate such clear definitions, it simplifies the comprehensibility of the subsequent text greatly!

Thank you for the comment!

**L157:** I'm not sure if I understand this sentence correctly. Do you mean that radiation fluxes significantly departure **locally** from areal-average fluxes?

Precisely. This sentence was clarified as follows (line 171):

*"Over mountainous regions, the local irradiance at the surface can exhibit significant departures from its areal-average value at spatial scales routinely resolved in ESMs due to the complexity of topography and surface properties."*

**L 162:** "represents the fraction of the sky dome visible from a target site" -> technically, this is incorrect. Compare e.g. with Helbig et al. (2009) (text next to Eq. 8) and Zakšek et al. (2011). The sky view factor definition of Dozier and Frew (1990) yields the fraction of

hemispherical radiation received under the assumption of isotropic radiation. The same is valid for the subsequent explanation of the terrain configuration factor $C_t$.

We agree with the comment. We have revised this sentence stating that the sky view factor used here is defined as the ratio of diffuse sky irradiance at a point to that on an unobstructed horizontal surface, in case of isotropic diffuse radiation.

We modified the sentence as follows (now at line 175):

*The terrain variables used to predict downward fluxes are i) the sky view factors $V_d$, which represents the ratio of diffuse sky irradiance at a point to that on an unobstructed horizontal surface, under the assumption of isotropic diffuse radiation (Dozier and Frew, 1990; Helbig et al., 2009; Zaksek et al., 2011), ii) the terrain configuration $C_t$ which quantifies the contribution to the irradiance at a point originated by reflections from surrounding mutually visible slopes; iii) the solar incident angle $\mu_i$ i.e., the angle between the direct solar beam and the normal to the surface. These terrain variables are derived from the Shuttle Radar Topography Mission (SRTM) dataset (Farr et al., 2007) high-resolution (90 m) terrain information.*

*In order to compute the sky view factor, we use the rapid procedure proposed by (Dozier and Frew, 1990) whereby the unobstructed fraction of sky hemisphere is approximated as*

$$V_d \simeq \frac{1}{2\pi} \int_0^{2\pi} \left[ cos\theta_s sin^2 H_\phi + sin\theta_s cos\left(\phi - \phi_s\right)\left(H_\phi - sinH_\phi cosH_\phi\right) \right] d\phi$$

*for a point with slope $\theta_s$, aspect $\phi_s$, and horizon angle $H_\phi$(i.e., angular distance between zenith and local horizon along the generic azimuth direction $\phi$). The terrain configuration, which quantifies the reflected radiation received by surrounding slopes in direct sight, can then be obtained as $C_t \simeq (1 + \cos\theta_s)/2 - V_d$.*

**L171:** Could you explain why you use this terrain configuration definition and not simply $C_t = 1.0 - V_d$ (compare e.g. with Chu et al., 2021)?

Thank you for pointing out this reference.

According to Dozier and Frew (1990) and e.g., Chen et al. (2006) the relation between sky view factor and terrain configuration factor can be approximated to (See eq. 9b in Dozier and Frew 1990, or eq. 6 in Chen et al, 2006):

$$C_t = \frac{1}{\pi} \int_0^{2\pi} \int_{H_\phi}^{\psi_\phi} \sin\theta[\cos\theta\cos\theta_s + \sin\theta\sin\theta_s\cos(\phi - \phi_s)]d\theta d\phi$$
$$\approx \frac{1 + \cos\theta_s}{2} - V_d$$

The relation $C_t$ = 1.0 – $V_d$ can be further obtained by assuming locally flat surface $(\cos\theta_s = 1)$ , but in general does not hold for a sloping surface.

**L174:** I would briefly introduce and explain the parameters $\mu_i$ and $\mu_0$ here.

We have added the following clarification at line 192:

*Here $\mu_i$ is the cosine of the solar incidence angle (i.e., angle between the incoming direct light beam and the normal to the land surface) while $\mu_0$ is the cosine of the solar zenith angle (i.e., the incidence angle with respect to a horizontal plane).*

**L194:** I'm a bit confused by these lines. It seems that you perform the clustering only for soil elements (also according to line 207; kp and kp + 2) and not for glaciers and lakes. What is the reason behind this? I guess glaciated areas and lakes can also have very variable topographic parameters (like e.g. sky view factor).

Yes, that is correct. The reason for this choice is that in the current GFDL land model configuration, glacier and lakes are treated separately from "soil" land. In the current work, we do cluster the non-glaciated component in hillslopes and inner clusters but indeed consider lakes and glaciers each as a single cluster characterized by their areal-average topographic properties.

However, we note that in our study domains glacier and lakes constitute just a small fraction of the total area. There is no reason why glacier and lakes could not also be subdivided in multiple tiles if they occupy a relevant portion of the domain. This would be directly comparable to what we have done here, since the only difference for the purpose of our study would be the reflectivity of the surface. Our results here would equally apply to the case in which lake and glaciers were also partitioned in multiple clusters, since for the purposes of this study these three land classes behave the same way. We now make this point clear in the manuscript:

Revised as follows (line 211):

*The land fraction of the study sites, which are chosen to represent in size a typical ESM grid cell, is first divided in a maximum of three components: soil, glacier, lake. The soil fraction is then subdivided into a set of tiles characterized by homogeneous terrain properties relevant*

*for capturing the effects of topography on radiative transfer. Additionally, lake and glacier areas, where present, are treated as individual separate tiles. We note that in the domains selected for this study lake and glacier areas constitute a small fraction of the total grid cell area. When applying the methodology to areas where glaciers cover a large fraction of the grid cell, it may be useful to also partition glacier areas in multiple clusters. This can be done following the same methodology described here, since for the purpose of radiation-terrain interactions the only relevant parameters would be the average albedo of each cluster (land or glacier). In our approach, the land clustering is based on 4 terrain variables: The normalized sky view factor, terrain configuration, $S_{sa}$ and $C_{sa}$.*

**L204:** It's not obvious to me why you apply the clustering a second time. Generally, to increase the comprehensibility of this section, it might be worth to extend the workflow diagram displayed in Fig. 4. One could show the classification of a certain domain (resolved for every single step).

*[see also following comment, and our response to major comment #1]* We agree that this point should be clarified in the text and will add a workflow diagram as suggested.

The main reason for the multi-level clustering is obtaining a flexible sub—grid structure able to be applied to other processes other that the topographic radiation correction pursued here. We discuss this in detail in the response to comment #1.

We have added a new section in the manuscript where we discuss this issue and the reasoning for the hierarchical clustering structure adopted. At line 206:

*In order to capture the spatial variability of radiative fluxes, here we employ a hierarchical multivariate clustering approach (HMC) which was recently introduced to study the role of heterogeneity in hydrological and land models (Chaney et al., 2016, 2018). Here we tailor HMC to the case of shortwave radiative fluxes by performing the land clustering based on terrain properties (namely $\mu_i, V_d, and\ C_t$) which are known to modulate the downwelling radiation over mountains as discussed in the previous section.*

*The land fraction of the study sites, which are chosen to represent in size a typical ESM grid cell, is first divided in a maximum of three components: soil, glacier, lake. The soil fraction is then subdivided into a set of tiles characterized by homogeneous terrain properties relevant for capturing the effects of topography on radiative transfer. Additionally, lake and glacier areas, where present, are treated as individual separate tiles. We note that in the domains selected for this study lake and glacier areas constitute a small fraction of the total grid cell area. When applying the methodology to areas where glaciers cover a large fraction of the grid cell, it may be useful to also partition glacier areas in multiple clusters. This can be done*

*following the same methodology described here, since for the purpose of radiation-terrain interactions the only relevant parameters would be the average albedo of each cluster (land or glacier). In our approach, the land clustering is based on 4 terrain variables: The normalized sky view factor, terrain configuration, $S_{sa}$, and $C_{sa}$. Note that these variables are independent of the sun's position. Once the direction of the incoming beam is given ($\phi_0$ and $\cos\theta_0$), average values of $S_{sa}$ and $C_{sa}$ over any given tile uniquely identify the solar incident angle for each point on the land surface by means of Eq. 4.*

We have updated the workflow in Figure 3 to show the steps used to construct hillslopes and clusters.

[Figure]

"Figure 3: Schematic representation of the land clustering workflow. Land elevation maps are used to compute the variables of interest (sky and terrain view factors, and functions of terrain slope and aspect) which are then used in the hierarchical clustering step, which yields a map of homogeneous land units used to parameterize radiation-topography interactions. The delineation of channel network, watersheds and hillslopes follows the approach developed by Chaney et al., (2018)."

**L207:** I'm still a bit puzzled – what is the motivation behind categorizing land surface based on hydrological properties? I don't see the connection to topography-radiation-processes.

[*See our response to the previous comment, and to major comment #1*]

**L298:** "reflected components are quite linear" -> for $f_{rdir}$, the deviations between MLR and RFR are quite substantial…

The sentence was indeed incorrect. The discrepancy observed for small solar zenith angles was described later separately. We have now corrected and rewritten the entire sentence which now reads (line 329):

*"As shown in Figure 7, the difference between RFR and MLR prediction skills is most relevant for the direct and direct-reflected flux components in the case of low solar zenith angles, and are also significant in the case of diffuse and coupled fluxes. In these two latter cases, the difference between RFR and MLR skills persist for all $\mu_0$ values and thus is not limited to cases where the sun is relatively low on the horizon, as is the case for direct and direct-reflected fluxes. The reflected-diffuse component is quite linear, as shown by the small difference between random forest and linear regressors predictive skills. For the direct-reflected flux this only happens for large enough values of the cosine of the solar zenith angle ($\mu_0 >$ 0.55)."*

**L 305:** "case in which…" -> I don't understand this part; there is probably something missing.

We agree, the sentence was unclear. We have revised it as follows at line 341:

*"In the case of larger spatial averaging scales and larger solar angles, the MLR describes the direct flux with great accuracy".*

**L316:** First of all, I'm confused about which region (East Alps vs. Peru) is the (in-)dependent domain. The caption of Fig. 7 does not agree with the statement here. Furthermore, I'm not convinced that results from RFR are not location dependent. Looking at Fig. 7, the RFR method consistently indicates a worse performance for the cross-validation domain than the MLR method. For me, this is an indication that obtained relations from the RFR simulation are very location-dependent and not easily transferable to other terrain geometries (i.e. the model is overfitted).

As stated in the caption of Figure 7, here *"The models were trained over the Peru site and tested over the same site (SS, continuous lines) and over the independent EastAlps site for cross validation (CV, dashed lines). ".* This was indeed correct. At line 316 we meant to say that we also did run the other configuration (switching the training and testing sites) and by doing so obtained similar results. We now make this clear at line 352:

*These out-of-sample results were obtained using training data from the Peru domain and testing data from the EastAlps domain. For completeness, we did also run the opposite*

*configuration (switching training and testing domains, in the online supporting material) finding similar results.*

We agree with the second part of this comment. The statistical models used "learns" from the input data and this is indeed the reason why the cross validation was performed. We think it is important to quantify the magnitude of this effect, which here we do by comparing in-sample and out-of-sample predictions. Of course, for applications we are interested in extrapolating the method to different regions. For extrapolation purposes, we agree that the in-sample performance is not representative. For this reason, the out-of-sample goodness of fit (which, while not being as good as the in-sample one, is not that bad) can be used to assess method performance. We also agree with the observation that RFR does appear to overfit the training data to some extent (exhibiting a larger difference between in- and out-of-sample cases with respect to MLR), and this is the reason why in the discussion we argue about the advantages of the linear model (MLR), despite some nonlinear behavior being observed for some of the flux components.

We have now expanded the discussion of this tradeoff at line 355:

*"In all cases in which predictive skills of MLR and RFR diverge, we observe that when comparing in-sample and out-of-sample performance the loss of predictive skill is larger in the case of RFR. This finding is not surprising. Given the additional model complexity of the RFR approach with respect to MLR, our analysis confirms that it is more prone to overfitting the calibration dataset. Once this overfitting tendency is accounted for, our analysis selects the MLR as model of choice since applications of the methodology do inevitably require extrapolation of the results to new domains. Based on these results, we generally recommend the adoption of the linear regression models at least for direct and reflected fluxes, given the good performance and model simplicity. Applications of RFR are in principle possible in ESMs and has been shown here to have good predictive performance for this specific problem. However, this comes at the cost of a lower interpretability and based on the present analysis here RFR is not the model of choice, given the limited increase in predictive skill with respect to MLR especially when tested in cross validation."*

**L388:** It would be interesting to see the results for these tests too. Maybe you could show them in the supplementary material.

The results of these tests were already reported in the paper. We have improved the description of these results in the paper to make sure this is clear for the reader. At line 428:

*"To further analyze the configuration of the tiling structure used, we also tested different tiling configurations obtained by fixing the number of characteristic hillslopes (k=5) and*

*varying the number of lower-level land units in each hillslope (p), or conversely varying k with p=5 fixed. Results from both these approaches are reported in Figures 12 and 13. We find that generally convergence is faster using a larger p, i.e., dividing each characteristic slope in a larger number of tiles as opposed to increasing the number of characteristic hillslopes. This is not surprising. However, differences are generally small, and therefore the model proposed appears flexible and can in principle be applied with tiling predefined in order to accommodate for other physical processes as well."*

**Typos, phrasing and stylistic comments**

**L124:** references not correctly rendered

Agree, revised as suggested.

**L153:** I was a bit confused by this line, it might be better to write something like: "The MC calculations were performed for three independent domains (Nepal, Peru, East Alps)…" (if that is what you mean)

The statement about 2 domains here was correct - we indeed run the Monte Carlo simulations for two domains only, as explained above due to the computational expense and the need to repeat our analysis for several solar angles. Three domains are then used in the clustering study to get additional data points to evaluate the effect of the number of tiles, but the third domain is not used to train and test the predictive model for the radiation correction terms. In this part of the analysis, the parameterization (trained and tested previously) is applied separately to both the high-resolution digital elevation maps, and to the sub—grid tiling schemes for varying number of clusters.

We have added the following in the manuscript (line 167):

*"While only two domains are used for MC calculations due to the computational expense of this procedure, an additional domain located in high mountain Asia (Figure 2) is used to further test the results of the spatial clustering over areas with different topographic features."*

And at line 385:

*"While the MC simulations were performed over two domains only, here we perform the clustering analysis over all three domains comparing predictions obtained by applying the 3D radiation corrections to the original high-resolution data with the same approach to the sub--grid tiling structures for a varying number of land clusters."*

**L157:** "determines" -> "determine"

Agree, revised as suggested.

**L162:** "represent" -> "represents"

Agree, revised as suggested.

**L166:** "in order to compute **the** sky view factor"

Agree, revised as suggested.

**L198:** "eq. 6" -> "Eq. 6"

Agree, revised as suggested.

**L215:** "if these **are** present in a given grid cell."

Agree, revised as suggested.

**L220:** "**the** is the indicator" -> "is the indicator"

Agree, revised as suggested.

**L263:** "eqns. (1)" -> "Eq. (1)"

Agree, revised as suggested.

**L273:** "angles compute based" -> "angles comput**ed** based"

Agree, revised as suggested.

**L273:** "simulation (5)" -> "simulation (Fig. 5)"?

Agree, revised as suggested.

**L290:** I would rewrite this to e.g.: "...larger than approximately 5 km the effect disappears."

Agree, revised as suggested.

**L302:** "case in which" -> "a case in which"

Agree, revised as suggested.

**Figures and Tables**

**Figure 2:** The colorbar labelling is erroneous – I guess it should be "Elevation [m a.s.l.]". The same is true for the upper-left panel in figure 3. Furthermore, the degree symbol is missing for the cardinal directions.

Indeed, the colorbar in this figure label refers to elevation in meters above mean sea level (m.s.l.). We do not believe it is erroneous as reported in our submission. As abbreviation, meters above sea level [m a.s.l.] as suggested by the reviewer would also be correct. We have now modified the caption explaining that we refer to elevation above mean sea level to make it clearer for the reader.

**Figure 4:** It seems from these panels (x/y-coordinates) that the MC model was run on a map projection. Could you specify the projection somewhere?

We agree with the comment. The projection used here is an equal-area Mollweide projection. We have now included this information in the caption of Figure 4 in the revised manuscript.

**Figure 7:** $\mu_0$ not correctly rendered in caption

Revised as suggested.

**New references**

Chu, Q., Yan, G., Qi, J., Mu, X., Li, L., Tong, Y., et al. (2021). Quantitative analysis of terrain reflected solar radiation in snow-covered mountains: A case study in Southeastern Tibetan Plateau. Journal of Geophysical Research: Atmospheres, 126, e2020JD034294. https://doi.org/10.1029/2020JD034294

Helbig, N., Löwe, H., & Lehning, M. (2009). Radiosity Approach for the Shortwave Surface Radiation Balance in Complex Terrain, Journal of the Atmospheric Sciences, 66(9), 2900-2912. https://doi.org/10.1175/2009JAS2940.1

Zakšek, K.; Oštir, K.; Kokalj, Å½. Sky-View Factor as a Relief Visualization Technique. Remote Sens. 2011, 3, 398-415. https://doi.org/10.3390/rs3020398

**References**

Lee, Wei-Liang, K. N. Liou, and Alex Hall. "Parameterization of solar fluxes over mountain surfaces for application to climate models." Journal of Geophysical Research: Atmospheres116.D1 (2011).

Lee, W-L., et al. "A global model simulation for 3-D radiative transfer impact on surface hydrology over the Sierra Nevada and Rocky Mountains." Atmospheric Chemistry and Physics15.10 (2015): 5405-5413.

Dozier, Jeff, and James Frew. "Rapid calculation of terrain parameters for radiation modeling from digital elevation data." IEEE Transactions on geoscience and remote sensing 28.5 (1990): 963-969.

Chaney, N. W., Van Huijgevoort, M. H., Shevliakova, E., Malyshev, S., Milly, P. C., Gauthier, P. P., & Sulman, B. N. (2018). Harnessing big data to rethink land heterogeneity in Earth system models. Hydrology and Earth System Sciences, 22(6), 3311-3330.

---

## Referee Report (RR1)

I would like to thank the authors for the additional effort they put in the revision of the manuscript. Please find below some additional final remarks (line numbers refer to the revised manuscript with marked changes):

**Title:** the model version changed from 4.2 to 4.1 – is that correct (i.e. intended)?

**L186:** I would remove the reference to Zakšek et al. (2011). They define the sky view factor simply as the fraction of the visible sky – in contrast to studies like Dozier and Frew (1990) and Helbig et al. (2009), which apply the correct definition of the sky view factor for radiation purposes.

**L196:** I'm still a bit puzzled by the approximation of the terrain configuration factor. Wouldn't the simple approximation $C_t = 1.0 - V_d$ be more accurate?

**L203:** I would explicitly state that the solar incidence angle is measured relative to the normal of the **horizontal** surface.

**L497:** "can be in principle be applied" → "can in principle be applied"

**L549:** I agree that a higher sub-grid tile structure would only be needed in areas with complex terrain. However, would the current model architecture be able to handle such an "unbalanced workload"? For instance: one computer cluster node would have to process a domain slice with complex terrain, while another one would process a domain slice with flat terrain. Would the former node not simply slow down the latter and thus determine overall run time? Or is your model able to distribute such an unbalanced workload evenly?

**Fig. 2:** The degree symbol is missing for the cardinal directions.

**Fig. 3:** I think I start to understand the splitting of ESM grid cells better but I still struggle to grasp the full details (after looking at Fig. 3 and re-reading Sect. 2.4). I think a reader would understand the splitting better if you illustrate the different stages by means of a single ESM grid cell. The smallest sub-grid units of the cell could then be colour-coded according to the current splitting/clustering stage. I.e. in a first stage, all units would have the same colour. In a second stage, you would have three colours (according to left side, right side and headwaters). And so on…

---

## Author Response (AR2)

**Response to reviewers - Minor revisions for manuscript EGUSPHERE-2022-770**

We wish to thank the editor and referees for useful comments on our work. In the following we report a point-by-point response to the comments (in blue).

The supplementary material code that was included as a .zip file with our original submission has now been moved to a dedicated Zenodo repository ( https://zenodo.org/record/7720281#%23.ZAvnArTMIpN ) Referred to in the manuscript and listed in the online assets.

**Referee # 1**

Thank the authors for their efforts to resolve my comments. I just have some minor comments for the figures:
1. Add the units for the x/y labels of latitude and longitude
2. Add the numbering for the sub-figures

We thank the reviewer for the attention devoted to our work. In the revised manuscript we have added latitude and longitude units to the map labels. We have also added numbering for the sub-plots of all figures with multiple sub-plots where this was missing (except in Figure 6 and 7, in which sub-figures already have row and column labels).

**Referee # 2**

I would like to thank the authors for the additional effort they put in the revision of the manuscript. Please find below some additional final remarks (line numbers refer to the revised manuscript with marked changes):

We thank the reviewer for the attention devoted to the manuscript.

**Title:** the model version changed from 4.2 to 4.1 – is that correct (i.e. intended)?

Yes, this is intended, as stated in our response to the previous round of reviews. Thank you for pointing this out. The methodology developed here can be applied to both model versions, but ESM 4.1 is the published version of reference (Dunne et al., 2020, cited in the paper) and the title of the manuscript now reflects the correct published model version. The LM4.2 is the version currently under active development at GFDL in but not the published reference version for this work.

**L186:** I would remove the reference to Zakšek et al. (2011). They define the sky view factor simply as the fraction of the visible sky – in contrast to studies like Dozier and Frew (1990) and Helbig et al. (2009), which apply the correct definition of the sky view factor for radiation purposes.

We agree and remove the citation.

**L196:** I'm still a bit puzzled by the approximation of the terrain configuration factor. Wouldn't the simple approximation Ct = 1.0 – Vd be more accurate?

According to Dozier and Frew (1990), page 965, for given S local slope:

$$C_t = \frac{1}{\pi} \int_0^{2\pi} \int_{H_\phi}^{\dot{V}_\bullet} \eta_v(\theta, \phi) \sin \theta \left[ \cos \theta \cos S \right.$$

$$\left. + \sin \theta \sin S \cos (\phi - A) \right] d\theta \, d\phi \qquad (9a)$$

$$C_t = \frac{1 + \cos(S)}{2} - V_d \qquad (9b)$$

*"Rigorous calculation of C, is difficult because it is nec
essary to consider every terrain facet visible from a point
to calculate $\eta_v$ [the anisotropy coefficient]. In contrast to the sky radiation, the iso
tropic assumption is unrealistic because considerable an
isotropy results from geometric effects, even if the sur
rounding terrain is a Lambertian reflector or a blackbody
emitter. We therefore note that Vd for an infinitely long
slope is (I + cos (S) ) /2, which leads to the approximation
in (9b)."*

The calculation proposed by Dozier and Frew 1990 which yields eq. the approximation (9b) reported here is obtained by integrating eq. (9a) between the horizon and the tangent to the local slope. The relation Ct = 1.0 – Vd is obtained from eq. (9b) if one then further assumes that the surface is locally flat, i.e., cos(S)=1. Therefore, we believe it is adequate to use eq. (9b) in this application.

**L203:** I would explicitly state that the solar incidence angle is measured relative to the normal of the **horizontal** surface.

This information is reported at line 192 of the revised manuscript:

*"Here μi is the cosine of the solar incidence angle (i.e., angle between the incoming direct light beam and the normal to the land surface) while μ0 is the cosine of the solar zenith angle (i.e., the incidence angle with respect to a horizontal plane). "*

**L497:** "can be in principle be applied" to "can in principle be applied"

Amended.

**L549:** I agree that a higher sub-grid tile structure would only be needed in areas with complex terrain. However, would the current model architecture be able to handle such an "unbalanced workload"? For instance: one computer cluster node would have to process a domain slice with complex terrain, while another one would process a domain slice with flat terrain. Would the former node not simply slow down the latter and thus determine overall run time? Or is your model able to distribute such an unbalanced workload evenly?

This is indeed a well-understood load imbalance issue, and the infrastructure used for running the GFDL ESM is already geared toward solving such problem. We have added the following clarification in the manuscript:

*"We note that in the case of the GFDL ESM, the model infrastructure is already suited for grids of this type, which can be characterized by an uneven number of sub--grid units in different grid cells depending on the local terrain properties. On the model start, the land grid cells can be distributed among available processors based on the estimated workload needed for each of the cells, assuming that computational cost is proportional to the number of sub--grid units. Therefore, the work per processor is roughly the same, and the imbalance resulting from this uneven grid structure is minimized. "*

**Fig. 2:** The degree symbol is missing for the cardinal directions.

A suggested we have now added the degree symbol and units to the map axis wherever needed.

**Fig. 3:** I think I start to understand the splitting of ESM grid cells better but I still struggle to grasp the full details (after looking at Fig. 3 and re-reading Sect. 2.4). I think a reader would understand the splitting better if you illustrate the different stages by means of a single ESM grid cell. The smallest sub-grid units of the cell could then be colour-coded according to the current splitting/clustering stage. I.e. in a first stage, all units would have the same colour. In a second stage, you would have three colours (according to left side, right side and headwaters). And so on...

We agree that this addition would clarify the procedure. We have revised Figure 3 as follows, showing the workflow of our methodology to partition a single grid cell in hillslopes and tiles:

[Figure]

In a first step, based on this set of variables, the land component of the domain is first divided in characteristic hillslope elements. These are obtained by first delineating catchments A conceptual summary of this clustering procedure is described in Figure 3. The digital elevation map (Figure 3A) is used to compute the drainage network (Figure 3E) necessary to partition the domain in basins (Figure 3B, 3C) based on a threshold area of $1 \times 10^5 \, \text{m}^2$ and by dividing each basin. Basins are in turn subdivided in hillslopes (Figure 3F) following (Chaney et al., 2018): Each basin is divided in up to three contiguous hillslope elements, corresponding to left side, right side, and headwaters.

Then, hillslope elements are aggregated in $k$ *"characteristic hillslopes"* via k-means clustering (MacQueen et al., 1967) in the 4-dimensional space of the variables $\tilde{V}_d$, $\tilde{C}_t$, $S_{sa}$, and $C_{sa}$; this enables to obtain land units characterized by similar radiation-topography interaction. Figure 3G shows the spatial distribution of these characteristic hillslope clusters for the case $k = 5$.

Then, each land unit so obtained Finally, each of these $k$ land units is further partitioned into $p$ sub-units by a second application of the k-means clustering algorithm based on the 4 variables $\tilde{V}_d$, $\tilde{C}_t$, $S_{sa}$, and $C_{sa}$. A conceptual summary In figure 3H we show the result of this procedure is described in Figure 3. which yields 25 tiles in this example.